# Refined CRISPR/Cas9 genome editing in the pea aphid uncovers the essential roles of *Laccase2* in overwintering egg adaptation

Shuji Shigenobu[1,2,3☯*], Shinichi Yoda[1☯], Sonoko Ohsawa[1], Miyuzu Suzuki[1]

**1** Laboratory of Evolutionary Genomics, Trans-Scale Biology Center, National Institute for Basic Biology, Okazaki, Japan, **2** Life Science Center for Survival Dynamics, Tsukuba Advanced Research Alliance (TARA), University of Tsukuba, Tsukuba, Japan, **3** Department of Basic Biology, School of Life Science, SOKENDAI (The Graduate University for Advanced Studies), Okazaki, Japan

☯ These authors contributed equally to this work.
* shige@nibb.ac.jp

## Abstract

The production of overwintering eggs is a critical adaptation for winter survival among many insects. Melanization contributes to eggshell pigmentation and hardening, consequently enhancing resistance to environmental stress. The complex life cycle of the pea aphid (*Acyrthosiphon pisum*), a model hemipteran insect with remarkable reproductive capacity, involves cyclical parthenogenesis. It enables the production of black overwintering eggs that undergo obligate diapause to survive under unfavorable conditions. Laccase2 (*Lac2*) is essential for cuticle sclerotization and pigmentation in other insects. We hypothesized that *Lac2* plays a critical role in aphid eggshell pigmentation and survival during diapause. To test the hypothesis, we used CRISPR/Cas9 ribonucleoprotein microinjections and a novel Direct Parental CRISPR (DIPA-CRISPR) method to knockout *Lac2*. In *Lac2* knockout (KO) crispants (G0), pigment-less eggs correlated with induced indel rates. Additionally, eggshell pigmentation was completely lost in homozygous *Lac2* knockouts, leading to embryonic lethality. Observation of late-stage embryos in KO diapause eggs suggested that lethality occurred during late embryogenesis or hatching. Furthermore, eggshell stiffness was significantly reduced in *Lac2* KOs, highlighting the role of this gene in eggshell hardening. Moreover, fungal growth was observed in KO eggs. These findings reveal the essential roles of *Lac2* in eggshell pigmentation, hardening, late embryonic development, hatching, and fungal protection, which are critical for pea aphid survival during overwintering diapause. This study also advances CRISPR/Cas9-mediated genome editing in pea aphids by addressing the challenges associated with their unique biology, including complex life cycles, obligatory diapause, bacterial endosymbiosis, inbreeding depression, and high nuclease activity. Our optimized protocol achieved efficient targeted mutagenesis and germline transmission, thereby generating stable KO lines. Additionally, we successfully applied DIPA-CRISPR to aphids

**Data availability statement:** The raw Illumina amplicon-seq reads were deposited in the DNA Data Bank of Japan (DDBJ) under the accession number PRJDB19177. The code used in this study is available on the GitHub repository (https://github.com/shigenobulab/24shuji-lac2crispr-paper).

**Funding:** This work was supported in part by JSPS KAKENHI Grant Numbers 24H00580, 20H00478, and 17H03717 to SS (https://www.jsps.go.jp), HHMI Janelia's Visiting Scientist Program (2013-2017) to SS, JSPS KAKENHI Grant Number 22K15169 to SY, and The Sumitomo Foundation Grant for Basic Science Research Projects (Grant Number 210448) to SY (http://www.sumitomo.or.jp/e/). The funders had no role in study design, data collection and analysis, decision to publish, or preparation of the manuscript.

**Competing interests:** The authors have declared that no competing interests exist.

by inducing mutations via adult oviparous female injections in fertilized eggs. These robust genome-editing protocols will facilitate functional studies in aphids, a key model for research on evolution, ecology, development, and agriculture.

## Author summary

Surviving harsh winters is a challenge for many insects, and the production of specialized overwintering eggs is a common adaptation strategy. These eggs are protected from cold, desiccation, and fungal infections by their hardened, pigmented shells. The pea aphid, a hemipteran insect with a complex life cycle, relies on these eggs to survive winter. Aphids alternate between asexual reproduction in the warmer months and sexual reproduction in the fall, producing overwintering eggs that remain dormant until spring. In this study, we explored the role of Laccase2 (*Lac2*), a key gene that contributes to the strengthening and darkening of insect exoskeletons. We disrupted *Lac2* in pea aphids using advanced genome-editing techniques, including a new approach called DIPA-CRISPR. This resulted in pigment-free eggs with weakened shells that were more prone to fungal infections and failed to hatch. This demonstrates that *Lac2* is essential for the survival of overwintering eggs. In addition, we refined the CRISPR/Cas9 genome editing methods for pea aphids, enabling efficient and precise genetic studies. These findings and the tools we developed can facilitate research on ecology, evolution, and pest control, while shedding light on how insects adapt to challenging environments.

## Introduction

Insects use various adaptation strategies to survive extreme environmental conditions, particularly in temperate regions, where seasonal changes require strategies for enduring cold winters. Among these, the production of overwintering eggs allows many species to survive during winter [1]. These eggs survive environmental stresses such as desiccation, low temperatures, and UV radiation. In addition, their resilience is often linked to structural changes in eggshells. Melanization, which involves melanin deposition, contributes to eggshell darkening and hardening [2,3]. This process likely enhances egg resistance to environmental stress. Although tanning reinforces eggshells through quinone-mediated protein-chitin cross-linking [4], the specific role of melanization in overwintering egg survival remains unclear.

Laccase2 (*Lac2*) is involved in cuticular melanization. This multi-copper oxidase (MCO) catalyzes the oxidation of catechols to orthoquinones, facilitating both pigmentation and sclerotization [5]. *Lac2* facilitates cuticular hardening in various insects, particularly during development, by enhancing the structural integrity of the cuticle [2,6]. Although this gene is primarily expressed in the epidermis and is essential for cuticle formation, its expression or function in overwintering eggs remains

unclear. Most studies have focused on its roles in larval and adult cuticles, with little attention given to its involvement in eggshell formation and embryogenesis, or its potential contribution to the durability of overwintering eggs.

Aphids are Hemipteran insects that serve as models for studies of insect-plant interactions, symbiosis, virus vectoring, and polyphenism [7–9]. They are characterized by a complex life cycle known as cyclical parthenogenesis, which alternates between asexual viviparous parthenogenetic generations (without laying eggs) during spring and summer, and a sexual generation consisting of oviparous females and males in the fall. This sexual generation produces overwintering eggs that can survive winter. In the pea aphid (*Acyrthosiphon pisum*), a well-established model species, the cream-colored eggs deposited by sexual females darken to black and then enter diapause lasting approximately 100 days. During diapause, they undergo slow development in a cold winter environment before hatching in spring [10,11]. However, the mechanisms that enable these eggs to survive the harsh winter conditions remain unclear. Given the known roles of the *Lac2* gene in other insects, particularly in the melanization and hardening of cuticular structures, we hypothesized that *Lac2* also facilitates the survival of aphid eggs during overwintering.

Clustered Regularly Interspaced Palindromic Repeats/CRISPR-associated protein 9 (CRISPR/Cas9) is a powerful tool for efficient site-directed genome editing. Since its identification as a bacterial immune defense mechanism (reviewed in Lander, 2016) [12], CRISPR/Cas9-mediated gene editing has been successfully applied to numerous organisms, from bacteria to mammals. Functional studies on insects have also increasingly employed CRISPR-mediated genome editing; notable examples include mosquitoes [13], *Tribolium* beetles [14], honeybees [15], ants [16,17], butterflies [18], and planthoppers [19]. Despite these successes, significant challenges remain when applying this method to the pea aphid (*A. pisum*) because of its unique biological characteristics, including its complex life cycle [10,20] and obligate endosymbiosis [21,22]. By addressing these issues, a couple of recent studies have successfully demonstrated the knockouts of *stylin-01* and *DDC* in pea aphids [23,24]. However, there remain opportunities for further improvement. For instance, post-diapause hatching rates, survival rates, and germline transmission efficiency could be enhanced. In addition, the full workflow spans more than seven months, making it relatively time- and labor-intensive. Therefore, the efficiency of obtaining mutants and reducing laborious procedures warrants further optimization to increase the reliability and scalability of these approaches. Such refinements of CRISPR/Cas9 workflows could facilitate the broader application of genome editing in pea aphids and enhance their utility as model organisms.

Direct Parental CRISPR (DIPA-CRISPR), an alternative approach to the traditional CRISPR/Cas9 gene editing, has recently emerged. Initially developed for species such as the German cockroach (*Blattella germanica*) and the red flour beetle (*Tribolium castaneum*), this technique bypasses the need for labor-intensive microinjection into eggs by introducing Cas9 ribonucleoproteins (RNPs) directly into adult females [25]. The Cas9/sgRNA complex is presumably incorporated into developing oocytes along with the yolk protein precursors, leading to heritable mutations. DIPA-CRISPR presents a significant advantage over conventional methods, especially for researchers who are less familiar with egg microinjection techniques, because it requires only simple adult injections. Given the technical challenges posed by aphid biology, including their small egg size, DIPA-CRISPR is a promising alternative for aphid genome editing.

In this study, we aimed to establish an efficient protocol for CRISPR/Cas9-mediated genome editing of pea aphids, including the use of DIPA-CRISPR, to elucidate the role of Lac2 in aphid eggshell pigmentation and survival. We hypothesized that *Lac2* plays an important role in the survival of overwintering eggs during diapause. Our optimized genome editing approach highlights the essential role of *Lac2* in the facilitation of seasonal adaptation within the complex life cycle of pea aphids.

## Materials and methods

### Insects

We used three *A. pisum* strains, namely, ApL, KNb, and 09003A, originally collected from Sapporo, Japan. These strains were gifts from Dr. Shin-ichi Akimoto (Hokkaido University). The characteristics of each strain are described in a previous

PLOS Genetics

study by Matsuda et al. [26]. Parthenogenetic aphid clones of the stock populations were maintained on vetch seedlings (*Vicia faba*) in growth chambers under long-day photoperiod conditions (16 h light:8 h dark cycle) at 16 °C.

## Overview of the CRISPR/Cas9-mediated knockout workflow in the pea aphid

We developed a non-homologous end-joining (NHEJ)-mediated knockout method for pea aphids using the CRISPR/Cas9 system. The workflow is illustrated in Fig 1. First, asexual reproduction was induced to obtain eggs for the CRISPR/Cas9 injection. Aphids that were maintained regularly in the laboratory during asexual viviparous reproduction were transitioned to sexual reproduction by shifting to short-day conditions for 1.5–2 months. To reduce inbreeding depression [26], we outbred two strains: ApL (female) × 0903A (male) and ApL (male) × KNb.

Oviparous females that mated with males were allowed to lay eggs for 24 h, and the RNP complex was injected into the eggs (0–24 h after egg laying (AEL)). This injection time window was chosen because primordial germ cells (PGCs) are cellularized at approximately 24–28 h AEL during embryogenesis [27] to facilitate efficient germline mutagenesis. Our

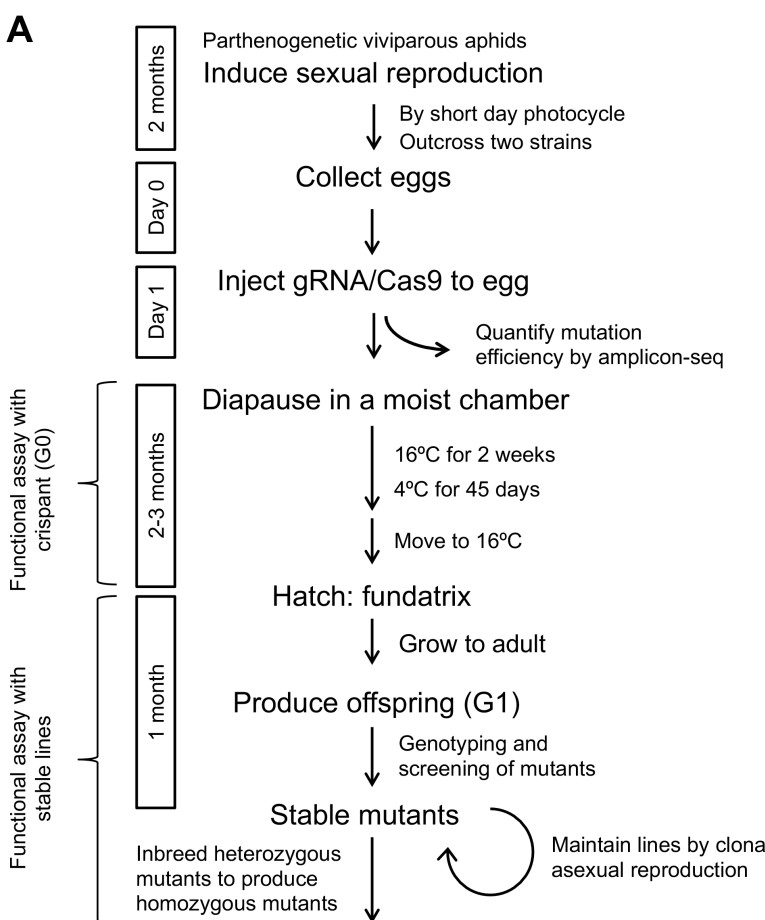

**Fig 1. An outline of the CRISPR/Cas9-mediated knockout workflow in the pea aphid.** Sexual reproduction is induced by rearing aphids under a short-day photoperiod (8 h light/ 16 h dark) at 15 °C, while the laboratory stock is maintained in a parthenogenetic, viviparous state under a long-day photoperiod (16 h light/ 8 h dark) at 16 °C. To mitigate inbreeding depression, two outcrossed combinations are used: ApL (female) × 0903A (male) and ApL (male) × KNb. A duplex of tracrRNA and target-specific crRNA (both components of the Alt-R CRISPR-Cas9 system by IDT) is incubated with Cas9 protein to form the CRISPR-Cas9 ribonucleoprotein (RNP) complex. The RNP complex was injected into the mid-to-posterior region of 0–24 h AEL (after egg laying) eggs, carefully avoiding disruption of the bacterial symbiont cluster located at the posterior pole.

injection targeted a region anteriorly adjacent to the cluster of endosymbiotic bacteria at the posterior end of the egg, where the germ plasm is localized and PGCs are formed. The germ plasm is invisible, but the symbiont cluster is visible in green. Although laid eggs initially exhibited an ivory color, healthy eggs turned black at 2–3 d AEL, which indicates survival without damage from the microinjection. A small subset of eggs was collected and subjected to amplicon sequencing to estimate the mutation efficiency in each batch of experiments.

Injected eggs were incubated at 16°C for 14 days and subsequently transferred to 4°C for 45 days in a moist chamber (see *Diapause and hatching* section below for details) to mimic natural overwintering diapause. Under natural winter conditions, pea aphids require a 100-day diapause period before hatching in spring [10]; we shortened the diapause period to 59 days to expedite mutant generation. At the end of the diapause period, eggs were transferred to 16°C waiting for hatching, which usually takes 0–10 days. Phenotypes were inspected in the hatched larvae (G0, founder generation; a.k.a. crispant).

The G0 larvae were grown to adulthood, and the fundatrices produced their offspring (G1) via viviparous parthenogenesis. Each G1 individual produced genetically identical clonal offspring, leading to the establishment of a stable line. Genotyping was performed using the RGEN-RFLP method, followed by Sanger sequencing or Illumina Amplicon-seq [28]. Mutant lines, confirmed by genotyping, were maintained as stable lines for further analyses.

### Induction of sexual generation and egg production

First-instar nymphs produced by viviparous aphids were reared on vetch seedlings at 15°C under short-day conditions (8 h light:16 h dark cycle) to induce sexual reproduction in asexual mode. The viviparous nymphs (F1) were grown to adulthood (approximately 2 weeks), transitioning into sexual-reproducing asexual females called sexuparae. Under short-day conditions, sexuparae produced males and oviparous females. To obtain fertilized eggs at 0–24 h AEL, males and oviparous females from different strains were placed together for 24 h in a plastic case containing vetch seedlings at a male-to-female ratio of 1:3 or 1:4. Eggs deposited on the seedlings were collected and injected as described below.

### Preparation of Cas9 RNP

Alt-R CRISPR-Cas9 crRNAs and custom tracrRNAs were purchased from Integrated DNA Technologies Inc. (IDT, IA, USA). The sequences of custom tracrRNAs are as follows. Lac2-4976AB:/AltR1/rUrUrG rGrCrA rCrGrG rArArU rCrCrA rCrCrA rGrArG rUrUrU rUrArG rArGrC rUrArU rGrCrU/AltR2/ (target sequence: TTGGCACGGAATCCACCAGA), and Lac2-4976BN:/AltR1/rGrCrU rGrUrG rArArU rCrGrU rArUrG rUrUrG rCrCrG rUrUrU rUrArG rArGrC rUrArU rGrCrU/AltR2/ (target sequence: GCTGTGAATCGTATGTTGCC). The tracrRNAs and crRNAs were annealed according to the manufacturer's instructions and stored at a concentration of 10 µM in a freezer until use. The annealed tracrRNA/crRNA (final concentration: 30 nM) was mixed with the Cas9 nuclease (Alt-R S.p. Cas9 Nuclease V3, IDT; cat no. 1081058) at a final concentration of approximately 30 nM and incubated at 25°C for 10 min, resulting in the formation of Cas9:gRNA complex at a 1:1 molar ratio.

DNA templates for the *in vitro* transcription (IVT) sgRNA were generated using a PCR-based method [29]. PCR reactions were set up with a 100 µM forward primer (CRISPRF-[target]), 100 µM reverse primer (sgRNAR), and PrimeSTAR HS premix (TaKaRa, Japan; cat no. R040A) in a 50 µl reaction volume. The cycling conditions were as follows: initial denaturation at 95°C for 10 s, followed by 35 cycles of 95°C for 10 s, 60°C for 5 s, and 72°C for 30 s. The oligos used were: sgRNAR: 5′-AAAAGCACCGACTCGGTGCCACTTTTTCAAGTTGATAACGGACTAGCCTTATTTTAACTTGCT ATTTCTAGCTCTAAAAC-3′; CRISPRF_Lac2–92: 5′-GAAATTAATACGACTCACTATAGGG CTGTGAATCGTATGTTGCC GTTTTAGAGCTAGAAATAGC-3′; and CRISPRF_Lac2–155: 5′-GAAATTAATACGACTCACTATAGGG ACGTCCTGAAC-CACATGCA GTTTTAGAGCTAGAAATAGC-3′. PCR products were purified using a QIAquick PCR Purification Kit (Qiagen, Hilden, Germany). The purified products were used as DNA templates for sgRNA synthesis using the MEGAshortscript T7

Transcription Kit (Thermo; cat# AM1354) according to the manufacturer's instructions. The sgRNAs were purified using the RNeasy Mini Kit (Qiagen) with a modified protocol that increased the ethanol volume from 250 μL to 700 μL to enable binding of smaller RNA fragments to the column. RNP complexes were prepared as described above using the synthesized sgRNAs.

## Microinjection into eggs

The injection needles were prepared from glass capillaries with filaments (1 × 90 mm, Narishige GD-1) using a needle puller (Narishige PC-10). The glass capillaries were treated with Sigmacote (SIGMA SL2) and dried completely in a fume hood. Next, the needles were pulled with a needle puller using the following settings: step = 1, heater level = 60.0, number of weights = 25 g × 2 + 100 g × 2). Before injection, the needle tips were gently scratched by touching the edge of the glass slide, which allowed the needle to open.

The eggs were attached to a microscope glass slide (Matsunami) using double-sided adhesive tape (NICHIBAN/ NW-R15), with the posterior end oriented toward the edge of the glass slide. The posterior pole was distinguished by the green color of the bacterial symbiont cluster. Microinjections were performed using a stereoscopic microscope (S8 APO; Leica, Wetzlar, Germany) equipped with a micromanipulator (MMO-220A and MN-4; Narishige, Tokyo, Japan) coupled to a FemtoJet express system (Eppendorf) with the following settings: injection pressure (Pi) = 900 hPa and compensation pressure (Pc) = 700 hPa. The RNP complexes, prepared as described in the *Preparation of Cas9 RNP* section, were injected into a region anteriorly adjacent to the symbiont cluster at the posterior end of the egg, where germ plasm is localized and PGCs form. Although the germ plasm was not directly visible, the symbiont cluster served as a marker because of its green coloration. The volume of RNP injected into the aphid eggs was estimated to be 0.01 – 0.05 μL.

## Diapause and hatching

After RNP injection, eggs were incubated on the slide glass in a moist chamber at 25°C for 1 h (This step is optional). The slides were then transferred to a moist chamber and incubated at 16°C for 2 weeks, followed by 45 days at 4°C to simulate diapause conditions.

The moist chamber was prepared as follows: a Ziploc container was used as the chamber base. A Kim Towel was moistened with tap water, wrung out to remove excess water, and placed at the bottom of the container. The glass slides containing injected eggs were arranged in a slide holder and placed on top of the damp Kim Towel. The container was then sealed with its lid to maintain a high-humidity environment throughout incubation. Under these conditions, the relative humidity was maintained to be nearly 100%. The chamber was regularly monitored to ensure the high humidity and to prevent desiccation and fungal infection. After diapause, the glass slide was transferred to a growth chamber at 16°C to mimic spring conditions and induce hatching.

Formalin fumigation was used to minimize fungal infections during extended diapause. Before RNP injection, eggs aligned on glass slides were placed in a Ziploc case containing Kimwipe paper moistened with formalin in an open 50-mL Falcon tube. The eggs were left in this setup for 5–10 min to ensure sterilization.

## *In vitro* Cas9 digestion

The newly designed tracrRNAs and crRNAs were evaluated through *in vitro* Cas9 digestion before their application in insects. The protocol provided by NEB (https://www.neb.com/ja-jp/protocols/2014/05/01/in-vitro-digestion-of-dna-with-cas9-nuclease-s-pyogenes-m0386) was followed, with minor modifications. Briefly, the reaction was assembled at room temperature as follows: 20 μl of nuclease-free water, 3 μl of NEBuffer r3.1, 3 μl of 300 nM sgRNA (final concentration: 30 nM), and 1 μl of 1 μM Cas9 Nuclease (final concentration: ~30 nM) (IDT), totaling 27 μl. The reaction components were pre-incubated for 10 min at 25°C. Next, 3 μl of 30 nM substrate DNA (final concentration: 3 nM) was added. PCR

products amplified from the pea aphid genome using primers ApLac2_F1 (5′-AGCTTGTCAAGTGTGCACAC-3′) and ApLac2_R306 (5′-CTGTTGGTGTCGAATTGGTATCTG-3′) served as substrate DNA. The reaction mixture was then incubated at 37°C for 15 min. To stop the reactions, 1 μl of Proteinase K was added to each sample, followed by a 10-min incubation at room temperature. Digested DNA fragments were purified using the MinElute Reaction Clean-up Kit (QIAGEN) and eluted in 10 μl of Buffer EB. Purified DNA fragments were subsequently analyzed using agarose gel electrophoresis.

## Genotyping through Amplicon-seq

Genomic DNA was extracted from the eggs or whole bodies of the aphids using either the DNeasy Blood and Tissue Kit (Qiagen) or a simplified method. For the simplified method, samples were homogenized using a Nippi-sterilized Bio-masher II (Funakoshi Co., Ltd., Tokyo, Japan) in Lysis Buffer A (10 mM Tris-HCl, pH 8.0; 0.1 M EDTA; 25 mM NaCl). The homogenate was treated with 400 μg/mL Proteinase K at 37°C for 1 h, followed by heat inactivation at 98°C for 2 min. Amplicon sequencing libraries were prepared according to the Illumina "16S Metagenomic Sequencing Library Preparation" (https://support.illumina.com/documents/documentation/chemistry_documentation/16s/16s-metagenomic-library-prep-guide-15044223-b.pdf). In the first round of PCR, the target region containing the gRNA-targeting site was amplified from individual genomic DNA using KAPA HiFi (Roche Diagnostics, Basel, Switzerland) with a primer pair containing barcode and overhang adaptor sequences. For the *Lac2* locus, we used the following primers: ApLac2_CT561F_IL: 5′-TCGTCGGCAGCGTCAGATGTGTATAAGAGACAGTACTTTGTGGAGCCACTGTCAG-3′ and ApLac2_CT810R_IL: 5′-GTCTCGTGGGCTCGGAGATGTGTATAAGAGACAGGAACGTGTTTCCCTCGTGAATC-3′. The PCR products were purified using AMPure XP (Beckman Coulter, Pasadena, CA, USA). A second round of PCR was performed using indexed primer sets to construct sequencing libraries using a Nextera XT Index Kit (Illumina, San Diego, CA, USA). The final libraries were purified using Ampure XP beads and sequenced on an Illumina MiSeq system in 150 × 150 paired-end format. Library construction and sequencing were conducted at the National Institute of Basic Biology (NIBB).

## Amplicon-seq data analysis

MiSeq reads were preprocessed using Cutadapt (ver. 1.13) to trim the adapter sequences and low-quality regions using the following parameters: -q 20 -O 8-minimum length, 25. The cleaned reads were mapped to the *A. pisum* reference genome (version Acyr_2.0; GCA_000142985.2) [8] using bowtie2 [30] with the end-to-end option. Alignments were converted to BAM format and visualized using IGV [31], focusing on the target sites. Indel calling was performed using mpileup in SAMtools [32] with the following parameters: -x -Q 13 -A-no-BAQ -d 1000000. The output was parsed with a custom Ruby script to generate summary statistics and was further analyzed using a custom R script for data visualization. The Shell, Ruby, and R scripts used in this analysis are available in our GitHub repository (https://github.com/shigenobulab/24shuji-lac2crispr-paper).

## Genotyping through RGEN-RFLP

RGEN-RFLP was performed as described previously [33], with some modifications. Briefly, 1 μM RNA duplex and 1 μM Cas9 nuclease (IDT) were mixed in 1×PBS and incubated at room temperature for 5 min to create the RNP complex. Next, 1.5 μl of the RNP complex and 4 ng of PCR-amplified products from the pea aphid genome (primer pairs: ApLac2_F1 and ApLac2_R306) were incubated at 37°C for 60 min in a 15 μl reaction containing NEB buffer 3. After the cleavage reaction, 0.5 μl of RNase A (100 mg/ml, #1007885; QIAGEN) was added, and the reaction mixture was incubated at 37°C for 30 min to degrade the RNA. Reactions were then stopped by heating at 65°C for 10 min. The products were purified using the MinElute Reaction Clean-up Kit (QIAGEN) and eluted in 10 μl of Buffer EB. The purified DNA fragments were analyzed using a TapeStation (Agilent).

## Reverse transcription-quantitative polymerase chain reaction (RT-qPCR)

RNA was purified using NucleoSpin RNA XS (Takara Bio Inc., Shiga, JP). *Laccase2* expression was evaluated through RT-qPCR using the One Step TB Green PrimeScript PLUS RT-PCR kit (Perfect Real Time) (Takara Bio) and the Roche LightCycler 96 real-time PCR instrument (Roche Applied Science, Penzberg, DE). RT-qPCR was initiated with a single round of reverse transcription at 42 °C for 5 min, followed by denaturation at 95 °C for 10 s. Subsequently, 45 cycles of PCR reactions were performed as follows: denaturation at 95 °C for 5 s, followed by annealing and extension at 60 °C for 20 s. Finally, a 3-step melting curve analysis was applied, starting with 95 °C for 10 s, followed by 65 °C for 60 s and 97 °C for 1 s. We used the ApLac2_F659 (5′-GCGAAGGAGACAAGGTCGTCA-3′) and ApLac2_R861 (5′-GGAATG GGCGTGCCAGAAATG-3′) primers to amplify *Lac2*, whereas the RpL7_F (5′-GCGCGCCGAGGCTTAT-3′) and RpL7_R (5′-CCGGATTTCTTTGCATTTCTTG-3′) primers [34] were used to amplify the Ribosomal protein L7 gene, serving as a reference for normalization.

## DIPA-CRISPR – Adult injection in oviparous aphids

Adult injections were administered to evaluate the baseline and optimized DIPA-CRISPR conditions in oviparous aphids. Initially, experiments were conducted to establish the effectiveness of Cas9 RNPs under standard conditions. Adult female aphids (KNb strain) were randomly selected and anesthetized on ice. The Cas9 RNP solution was injected into the hemocoel of each aphid using a FemtoJet 4i (Eppendorf) and a finely pulled glass capillary needle until the cauda visibly swelled. The injection solution, targeting the *Lac2* locus (5′-GCTGTGAATCGTATGTTGCC-3′) (Lac2-4976BN), was prepared by mixing 1.8 μL of 10 μM crRNA/tracrRNA duplex and 0.3 μL of 10 μg/μL Cas9 protein, resulting in final concentrations of 8.57 μM and 1.43 μg/μL, respectively. After injection, the females recovered at 16°C and were placed in plastic cases (cat# 02270c; Sanplatec) with two seedlings each. Males (ApL strain) were added at a 2:1 female-to-male ratio to ensure mating.

We conducted two sets of experiments to systematically evaluate the effectiveness of the Cas9 RNP injections in oviparous aphids. In the first set, we injected nine females and collected their eggs over one week following the injection, with the day after the injection designated as day 1. No eggs were collected on day 1. We collected eight eggs for genomic DNA extraction from days 2–7. In the second set of experiments, we injected six females and collected eggs at 2-day intervals to monitor temporal changes in indel rates. Specifically, six, seven, and twelve eggs were collected on Days 2–3, 4–5, and 6–7, respectively. The eggs were pooled separately for genomic DNA extraction. Pooled genomic DNA samples from each interval were subjected to amplicon sequencing to assess indel rates using the same amplicon-seq protocol previously described for egg microinjection analysis.

To optimize DIPA-CRISPR in oviparous aphids, we investigated three variables: mating time relative to the injection, age of the adult female at the time of injection, and the injection site. The mating timing was varied by allowing some females to mate before injection and others to mate immediately after injection. The adult females were divided into age groups from days 0 (day of adult emergence) to 8. Each group underwent the same injection procedure described in this section. We tested injection sites at the base of the third thoracic leg and on the dorsal or ventral sides of the abdomen, excluding the midline. Eggs were collected at 2-day intervals for eight days post-injection, individually processed for genomic DNA extraction, and subjected to amplicon sequencing to determine indel rates and assess genome editing efficiency. More than ten females were injected under each condition.

As a control, we injected Cas9 RNPs targeting the *BCR3* locus (5′-GCAACCACTGTAACATACGG-3′) using the same procedure as for the *Lac2*-targeted experiments. *BCR3* is specifically expressed in bacteriocytes, specialized cells involved in endosymbiosis with *Buchnera* [35]. The *BCR3*-targeting sgRNA exhibits high cutting efficiency, comparable to that of the *Lac2*-targeting sgRNA, and successfully generated knockout lines via egg microinjection. Results from these control experiments were compared with those from the *Lac2*-targeted experiments to evaluate the specificity and

effectiveness of DIPA-CRISPR in oviparous aphids. The same PCR primers were used for amplicon sequencing as those in the *Lac2*-targeted experiments to quantify potential off-target cleavage and background noise at the *Lac2* locus under identical conditions. This control experiment was conducted using four biological replicates.

### Immunohistochemistry

Ovaries were dissected from oviparous females 24 h post-Cas9 RNP injection and fixed in 4% paraformaldehyde in 1×PBS for 30 min. The ovaries were washed twice with PAT3 (0.5% Triton X-100/0.5% BSA in 1×PBS) and blocked with 3% goat serum in PAT3 (3% NGS/PAT3) for 3 h. Samples were then incubated overnight at 4°C with a His-tagged mouse monoclonal antibody (cat#70796; Novagen) diluted to 1:1000 in 3% NGS/PAT3. After three washes with PAT3, the ovaries were incubated overnight at 4°C with Alexa Fluor 647 goat anti-mouse IgG (cat#A21236; Thermo Fisher Scientific) diluted to 1:1000 in 3% NGS/PAT3. Next, the ovaries were washed three times with PAT3, counterstained with DAPI (1:1000 in 1×PBS), and washed three more times with PAT3. Finally, the samples were mounted in Vectashield Antifade Mounting Medium (cat#H-1000; Vector Laboratories) and analyzed using a Zeiss LSM 980 with an Airyscan 2 laser scanning confocal microscope. 3D image data were reconstructed from Z-stack images using ZEN software (Carl Zeiss Microscopy, Germany).

### Scanning electron microscopy (SEM)

SEM images of the eggshell surfaces were acquired using a Keyence VHX-D510 digital microscope. Egg samples collected within one month of laying were observed without pretreatment or coating. The eggs were then placed on a microscope stage and imaged under low-vacuum conditions to reduce the risk of sample charging. The microscope settings, including magnification and lighting, were optimized to capture detailed surface structures while maintaining sample integrity.

### Atomic force microscopy (AFM)

Young's modulus measurements were conducted using an AFM (CellHesion 200; Bruker, JPK Instruments AG, Germany) integrated with an Axio Zoom V.16 system (Zeiss). The Rtespa-150 cantilever (Bruker) (tip radius = 8 nm, length = 125 μm, and width = 35 μm) was used for the measurements. The aphid eggs were immobilized on glass slides using a manually applied layer of epoxy adhesive (Cemedine High Super 30; CEMEDINE Co., Ltd., Tokyo, Japan). Force curves of each eggshell were recorded using AFM, and the Young's modulus was calculated using the JPK data processing software (Bruker). The measurement parameters were as follows: setpoint = 30 nN, extension speed = 5 μm/s, sample rate = 1000 Hz, and measured spring constant range = 2.74–2.87 N/m.

### Statistical analysis

All statistical analyses were performed using R version 4.1.1. Data visualization was conducted with the ggplot2 package (ver. 3.5.1) in R. To assess differences in eggshell hardness, one-tailed Welch's t-tests were applied to the dataset of Young's modulus values.

### Phylogenetic analysis

The protein sequences were retrieved from the OrthoDB database (https://www.orthodb.org/) [36]. Orthologous Laccase genes were searched at both the Insecta and Arthropoda taxonomic levels using the text query "laccase." The search yielded 27 full-length amino acid sequences at the Insecta level and 55 sequences at the Arthropoda level from 14 species: *Acyrthosiphon pisum, Myzus persicae, Aphis gossypii, Rhopalosiphum maidis, Bemisia tabaci, Zootermopsis nevadensis, Drosophila melanogaster, Anopheles gambiae, Culex pipiens, Tribolium castaneum, Apis mellifera, Bombyx*

*mori, Papilio polytes,* and *Manduca sexta*. At the Arthropoda level, in addition to Laccase genes, multicopper oxidase-related proteins (MCORPs) were also retrieved. To identify MCORPs, we used the amino acid sequence previously reported as an MCORP in *T. castaneum* (NCBI accession: KJ500311) [37] as a query in OrthoDB. This search identified the *T. castaneum* sequence 7070_0:002c8d, which is identical to 070_0:002c8d, as the corresponding MCORP. From the same 14 species listed above, 16 additional full-length MCORP amino acid sequences were retrieved. The L-ascorbate oxidase gene from *Hypsibius exemplaris* (a tardigrade species) was used as an outgroup. A complete list of sequences is shown in S1 Table.

Multiple sequence alignments of 28 and 56 sequences from the Insecta and Arthropoda taxonomic levels, respectively, were performed using MUSCLE with default parameters. Phylogenetic trees were constructed using the maximum likelihood (ML) method with the LG model (G+I) for amino acid substitutions and partial deletions (95% site coverage cutoff). The best-fit model was selected using the MEGA model selection tool in MEGA11 [38]. Bootstrap analysis with 1,000 replicates was used to assess branch support for the Insecta dataset, and 500 replicates were used for the Arthropoda dataset. Phylogenetic trees were generated and visualized using MEGA11 software.

### Motif analysis and phylogenetic alignment of Lac2 from *A. pisum*

Domain analysis of the full-length Lac2 amino acid sequence from *A. pisum* was performed using InterPro (version 101.0) and three Cu-oxidase domains (accession numbers PF00394, PF07731, and PF07732), consistent with previous findings in other insect species [39]. Signal peptide predictions were performed using SignalP (version 6.0). Multiple alignments of Lac2 sequences were performed using MUSCLE with default parameters. The Lac2 sequences were extracted from a larger dataset used for constructing the ML tree (see "Phylogenetic analysis" in Materials and Methods section). A cysteine-rich region was identified based on this alignment and previous findings [40]. The multicopper oxidase signature regions were further extracted using PROSITE (accession numbers PS00079 and PS00080).

## Results

### Identification and phylogenetic position of *Lac2* in the pea aphid

*Lac2* encodes phenol oxidases and is critical for cuticle pigmentation and sclerotization in various insects, such as *Tribolium castaneum* [2], western corn rootworm [41], crickets [42,43], and stinkbugs [44]. A candidate *A. pisum Lac2* gene, identified via a BLAST search using *T. castaneum Lac2* (Gene ID: 641461) as a query, was annotated as *laccase-5* (NCBI Gene ID: 100164049). We used the OrthoDB (v12.0) ortholog database to verify its ortholog and correct annotation [36]. The orthogroup "Group 1603365at50557" at Insecta level included 787 genes in 707 species, representing the *Lac2* orthogroup. This group contains a single *A. pisum* gene (GeneID: 100164049), along with well-characterized *Lac2* homologs from *T. castaneum* (GeneID: 641461) [2] and *M. sexta* (GeneID: 115442328) [40]. Thus, we confirmed that the *A. pisum* gene model (NCBI Gene ID: 100164049; XP_001950788.1; UniProt: A0A8R2A7M0) is a *Lac2* ortholog. The database suggests that other aphid species, such as the corn aphid (*R. maidis*), the cotton aphid (*A. gossypi*), the green peach aphid (*M. persicae*), also contain a single copy of *Lac2* in each genome. *Lac2* gene is located on autosome 3 and consists of 12 exons in *A. pisum* (Fig 2A).

Multiple alignments of the amino acid sequences of *Lac2* orthologs from 13 insect species demonstrated evolutionary conservation of this gene across insects (Fig 2B). To further characterize the domains and motifs in *A. pisum* Lac2, we conducted a domain search using InterPro, which identified three well-defined Cu-oxidase domains (Pfam: PF00394, PF07731, and PF07732). SignalP analysis predicted a signal peptide at the N-terminus, indicating that Lac2 was likely to be secreted into the extracellular space, which is consistent with its expected role in the cuticle where phenol oxidase activity is required for sclerotization and melanization. Furthermore, a cysteine-rich region associated with the carbohydrate-binding domain [40] and a highly conserved multicopper oxidase signature sequence [Prosites PS00079 and PS00080] were identified (Fig 2B). These conserved features across *Lac2* sequences from various insect species

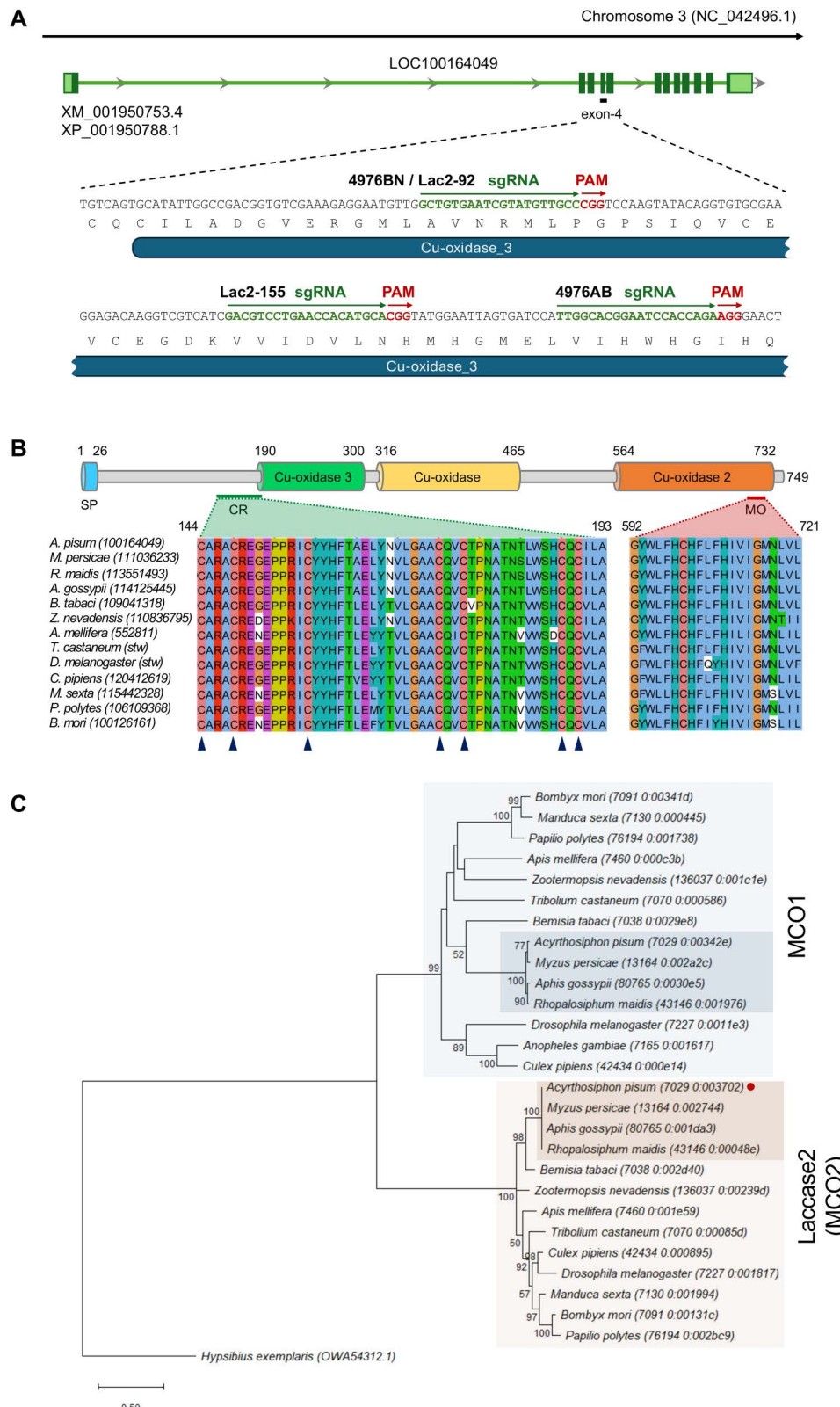

**Fig 2. The structure and phylogenetic context of *Lac2* in *A. pisum*.** (A) The *Lac2* locus in *A. pisum* is located on chromosome 3, as determined based on NCBI Gene annotation (LOC100164049). Exons are depicted in dark green for protein-coding regions and in light green for UTRs. A close-up

view of exon 4 highlights the target sites for CRISPR/Cas9-mediated gene knockout, with the designed sgRNAs and corresponding PAM sequences indicated. sgRNAs 4976BN and 4976AB were synthesized using the Alt-R CRISPR-Cas9 system (IDT), while Lac2-92 and Lac2-155 were prepared via in vitro transcription. Notably, 4976BN and Lac2-92 target the same genomic position. The position of the Cu-oxidase_3 domain (PF07732) within *Lac2* is also marked. (B) Domain architecture of the *A. pisum* Lac2 protein. The key domains are labeled, and boundaries are highlighted. Below the domain structure, cysteine-rich regions (CR) and multicopper oxidase signatures (MO) from *Lac2* orthologs in multiple insect species are shown. Arrowheads below the sequences indicate the cysteine residues conserved across all Lac2 proteins. A signal peptide (SP) is detected at the N-terminus. The sequence accession numbers are in parentheses. (C) Molecular phylogenetic tree of the full-length sequences of laccase2 (MCO2) and MCO1. A tree was constructed using the maximum likelihood (ML) method and the LG model. Branch lengths were proportional to sequence divergence, and the tree was rooted in L-ascorbate oxidase from *Hypsibius exemplaris*, a species of tardigrade. Bootstrap values (1,000 replicates) greater than 50% are shown at the nodes. The scale bar represents the number of substitutions per site. In total, 28 sequences were analyzed, with in-group sequences obtained from OrthoDB (https://www.orthodb.org/); their accession numbers are listed.

underscore the functional importance of these motifs in the enzymatic activities related to melanization and cuticle formation. Structural conservation also suggests that aphid *Lac2* retains molecular and enzymatic functions similar to those of *Lac2* orthologs in other insects.

In our initial phylogenetic analysis, we retrieved 27 Laccase-related sequences from OrthoDB at the Insecta taxonomic level. Most insects, including the four aphid species analyzed, possessed two Laccase genes that clustered into two distinct clades corresponding to *MCO1* (*Lac1*) and *Lac2*. Clustering showed strong support with high sequence conservation within these clades among aphids (bootstrap value = 100) (Fig 2C). Next, to expand the phylogenetic context of our analysis, we retrieved 71 MCO-related sequences at the broader Arthropoda level from OrthoDB, including not only Laccase genes but also MCORP sequences. Representative MCORPs were identified using the previously reported MCORP protein from *Tribolium castaneum* (NCBI accession: KJ500311 [37]) as a query. The L-ascorbate oxidase gene from the tardigrade *Hypsibius exemplaris* was used as an outgroup. The phylogenetic tree (S1 Fig) showed that the four aphid species consistently retain three distinct MCO gene groups: MCO1, Lac2, and MCORP. In contrast, other insect lineages such as *Culex pipiens* exhibit extensive lineage-specific gene expansions, with up to seven MCO genes spanning multiple clades. In aphids, however, MCO-related sequences were confined to their respective clades without evidence of recent gene duplications. This conserved MCO-related gene repertoire in aphids suggests a lack of order-level diversification and highlights the evolutionary stability and likely functional significance of these genes within the aphid lineage.

## Expression of *Lac2* during egg pigmentation and post-embryonic development in *A. pisum*

Eggs laid by sexual *A. pisum* females undergo a distinct color transition, initially appearing cream before gradually turning dull green and eventually darkening to black in several days [11] (Fig 3A). As *Lac2* encodes phenol oxidase, an enzyme involved in pigmentation in other insects, we hypothesized that it is critical for pigmentation during pea aphid egg development. To explore this hypothesis, we measured the temporal expression levels of *Lac2* transcripts in wild-type aphid embryos using qRT-PCR (Fig 3A).

Our analyses revealed a sharp increase in *Lac2* expression coinciding with the onset of egg pigmentation. The results showed a low baseline expression of *Lac2* before 20 h after egg laying (AEL), coinciding with the early cream-colored stage of the egg. *Lac2* expression began to increase sharply at approximately 30–40 h AEL, corresponding to the onset of green pigmentation in the eggs. It peaked between 40 and 50 h AEL, when the eggs displayed a dull green color. Thereafter, *Lac2* expression remained elevated but gradually declined as the eggs darkened to black by 70 h AEL. The timing of *Lac2* expression closely mirrored the progression of pigmentation, suggesting its involvement in the enzymatic processes that drive egg coloration. The peak in *Lac2* expression coincided with the most pronounced changes in pigmentation, further demonstrating its role in the early developmental stages of egg pigmentation.

We further examined the later stages of embryogenesis and post-embryonic development across various morphs, including viviparous females, oviparous females, fundatrix, and males, using qRT-PCR (Fig 3B). After an initial peak in *Lac2* expression during early egg development, expression was almost undetectable in one-week-old eggs. However, we

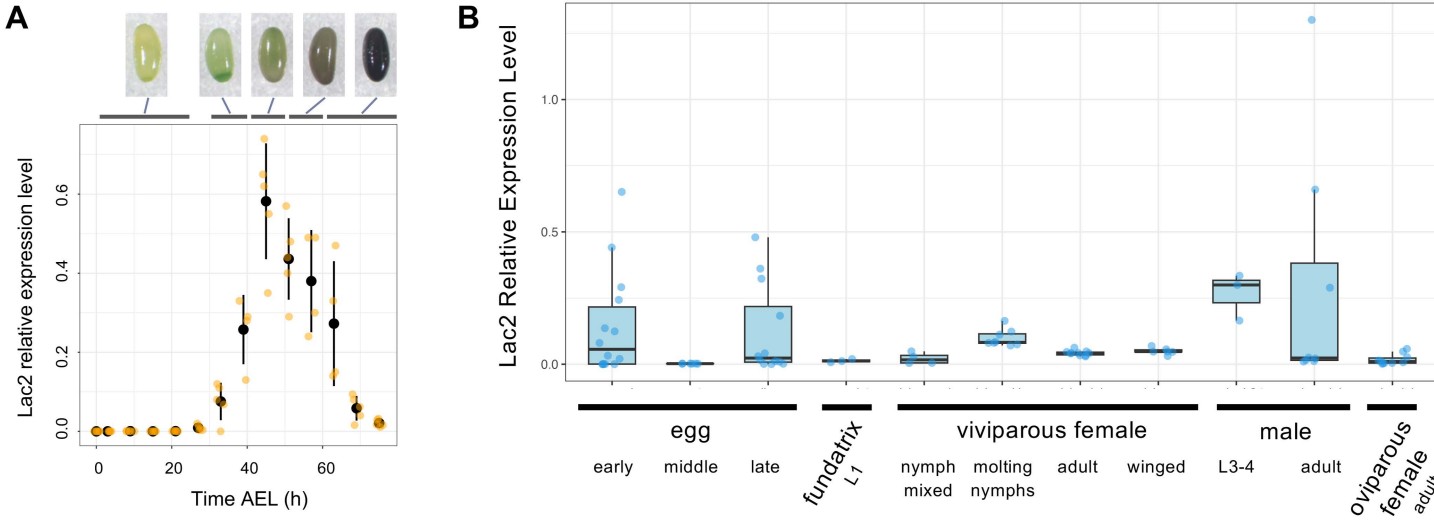

**Fig 3. Expression of *Lac2* mRNA.** (A) Temporal expression pattern of the *Lac2* transcript in early egg development, as analyzed using quantitative RT-PCR. The expression levels (Y-axis) are presented as values normalized to the house-keeping gene *rp49*. Data are plotted as means (black circles) and standard deviation (black lines). Individual data points are plotted as orange dots. The accompanying images depict eggs at five developmental stages (<24 h, 30– 40 h, 40– 50 h, 50– 60 h, and >70 h after AEL (after egg laying), illustrating the progressive pigmentation of the eggs. (B) *Lac2* transcript expression levels across various morphs and developmental stages, as analyzed using qRT-PCR. On the x-axis, "early" represents 0–72 h after egg laying, corresponding to data in (A); "middle" represents 1-week AEL; and "late" represents eggs at 1.5–2.0 months of AEL following diapause. "Fundatrix L1" indicates the L1 larva that hatches from overwintering eggs. For viviparous females, samples are further categorized as nymphs (L1–L4 stages mixed), molting, adult, and winged morphs. Winged viviparous morphs were induced under crowded conditions. In all cases, whole bodies or whole eggs were homogenized for qRT-PCR quantification. Expression levels are normalized to the housekeeping gene *rp49*.

observed a resurgence in *Lac2* expression in some eggs during the late diapause period before hatching (Fig 3B). Among the distinct morphs, males in the L3 and L4 nymph stages showed relatively high *Lac2* expression, whereas the fundatrix and oviparous females showed low *Lac2* expression. In viviparous nymphs, *Lac2* expression was elevated in aphids undergoing molting or in the pre- and post-molting stages, which is consistent with its reported role in cuticle formation in other insects.

## Optimization of CRISPR/Cas9-mediated knockout protocol in the pea aphid

To elucidate the role of *Lac2* in pea aphids, we knocked out the *Lac2* locus via the non-homologous end joining (NHEJ) pathway using CRISPR/Cas9-mediated genome editing. We tailored a CRISPR/Cas9-mediated knockout workflow to pea aphids, as outlined in Fig 1 to mediate the challenges associated with applying this method in aphids. While the full procedural details are provided in the Materials and Methods section, a summary of the key challenges and our solutions is provided below.

*Complex life cycle:* To ensure a constant supply of eggs, we established a workflow using multiple incubators with varying conditions that mimicked summer (16 h light: 8h dark cycle) and winter (8 h light: 16 h dark cycle), allowing the transfer of aphids to induce sexual reproduction.

*Inbreeding depression*: Our preliminary tests revealed a severe inbreeding depression in pea aphids [26]. Therefore, we collected nine aphid lines from various host plants (alfalfa *Medicago sativa*, red clover *Trifolium pratense*, *Vicia cracca*, and *Melilotus* spp.) and identified pairs that exhibited high hatching rates and fertility. Based on these results, we selected two outbred combinations, ApL (female) × 0903A (male) and ApL (male) × KNb, for use in subsequent experiments [26].

*Bacterial endosymbiosis*: Aphids harbor *Buchnera aphidicola*, an obligatory symbiont that is essential for their survival and is localized at the posterior pole of the egg. To avoid damaging these symbionts, we targeted our CRISPR/Cas9 injections to a region adjacent to the cluster of *Buchnera* at the posterior end of the egg, where the germ plasm is located and primordial germ cells (PGCs) are formed.

*High nuclease activity*: The internal environment of aphids is rich in RNases [45], which may inhibit efficient CRISPR/ Cas9 genome editing. We used the Alt-R CRISPR system (developed by IDT) to mediate this problem. This commercial tracrRNA:crRNA system includes 2′-O-methyl and phosphorothioate modifications on the ends of tracrRNA, as well as proprietary chemical modifications to crRNA. This increases resistance to nucleases and reduces the innate immune response (https://sg.idtdna.com/pages/products/crispr-genome-editing/alt-r-crispr-cas9-system). A comparison of chemically synthesized gRNAs (Alt-R) and gRNAs synthesized via IVT revealed a gRNAs (Alt-R)-induced significant improvement in indel efficiency (Fig 4A; see IVT gRNA-mediated CRISPR/Cas9 genome editing in S1 Text).

*Diapause*: Fertilized eggs must undergo a 3-month diapause in a winter-like environment before hatching [10], which presents a significant challenge for maintaining healthy eggs in the laboratory. Therefore, we developed a moist chamber system to maintain appropriate humidity levels and prevent desiccation. In addition, formalin was administered to prevent fungal infections.

With this optimized protocol, we targeted to knock out *Lac2* in the pea aphid to elucidate its function.

## CRISPR-induced mutations at the Laccase2 locus

**crRNA design for knockout and mutation efficiency.** We designed a transactivating CRISPR RNA (tracrRNA) targeting exon-4 or exon-5 to disrupt a conserved oxidase domain (Pfam:PF07732:Cu-oxidase_3; Multicopper oxidase) (Fig 2A). Chemically synthesized tracrRNAs and CRISPR RNA (crRNA) obtained from IDT were used to form the CRISPR-Cas9 ribonucleoprotein complex. Two crRNAs (Lac2-4976BN and Lac2-4976AB) were designed and the complex was injected into aphid eggs. Of the 1137 eggs injected, 585 (51.5%) survived. We then randomly selected a subset of the surviving eggs (5–37 days AEL) and conducted amplicon-seq analysis individually for 64 eggs.

Amplicon-seq analysis revealed indels in 49 of the 64 eggs (76.6%), indicating a high mutation efficiency. Indel ratios, which correspond to the population of cells harboring mutations in each mosaic knockout egg, varied among the eggs, with Lac2-4976BN inducing indels at an average rate of 29.1%, ranging from 1.6% to 86.6%, and Lac2-4976AB showing an average indel rate of 17.5%, ranging from 1.4% to 63.4% (Fig 4A). These results demonstrate effective gene disruption using both crRNAs. Furthermore, a pattern of elevated indels was observed with a strong peak at 3 bp 5' of the beginning of the PAM, the position at which Cas9 makes a double-strand break (Fig 4B). This indicates a precise targeted mutagenesis.

**Crispant phenotypic analysis.** We observed the *Lac2* mutation phenotype in the founder generation (crispants). Although the healthy aphid eggs turned from green to black, three eggs exhibited a lighter color (Fig 4C). These eggs failed to hatch after diapause. We conducted deep-sequencing analyses to investigate the effects of CRISPR-Cas9 mutagenesis. Amplicon-seq data demonstrated that the genomes of these crispant eggs were mutated at the targeted Cas9 site, with high indel rates observed (54.4–71.6%) (Fig 4D). The magnitude of pigmentation of crispant eggs was negatively correlated with the indel proportion (Fig 4C and 4D). The qRT-PCR assay of *Lac2* transcripts in wild-type aphids revealed that gene expression was coincident with egg pigmentation. Moreover, a sharp increase was observed in *Lac2* expression at the onset of pigmentation (Fig 3A). Thus, we concluded that the attenuated pigmentation observed in crispant eggs was the loss-of-function phenotype of *Lac2*.

**Germline transmission of mutant alleles.** We investigated whether somatic mutagenesis detected in adults reared from injected embryos (G0 animals) resulted in the transmission of stable mutant alleles to their offspring (G1 animals) through the germline. We focused on Lac2-4976BN crRNAs because this crRNA showed a higher somatic mutation rate in the crispant analysis (Fig 4A), with an expectation of a better chance of obtaining germline mutants. Embryos

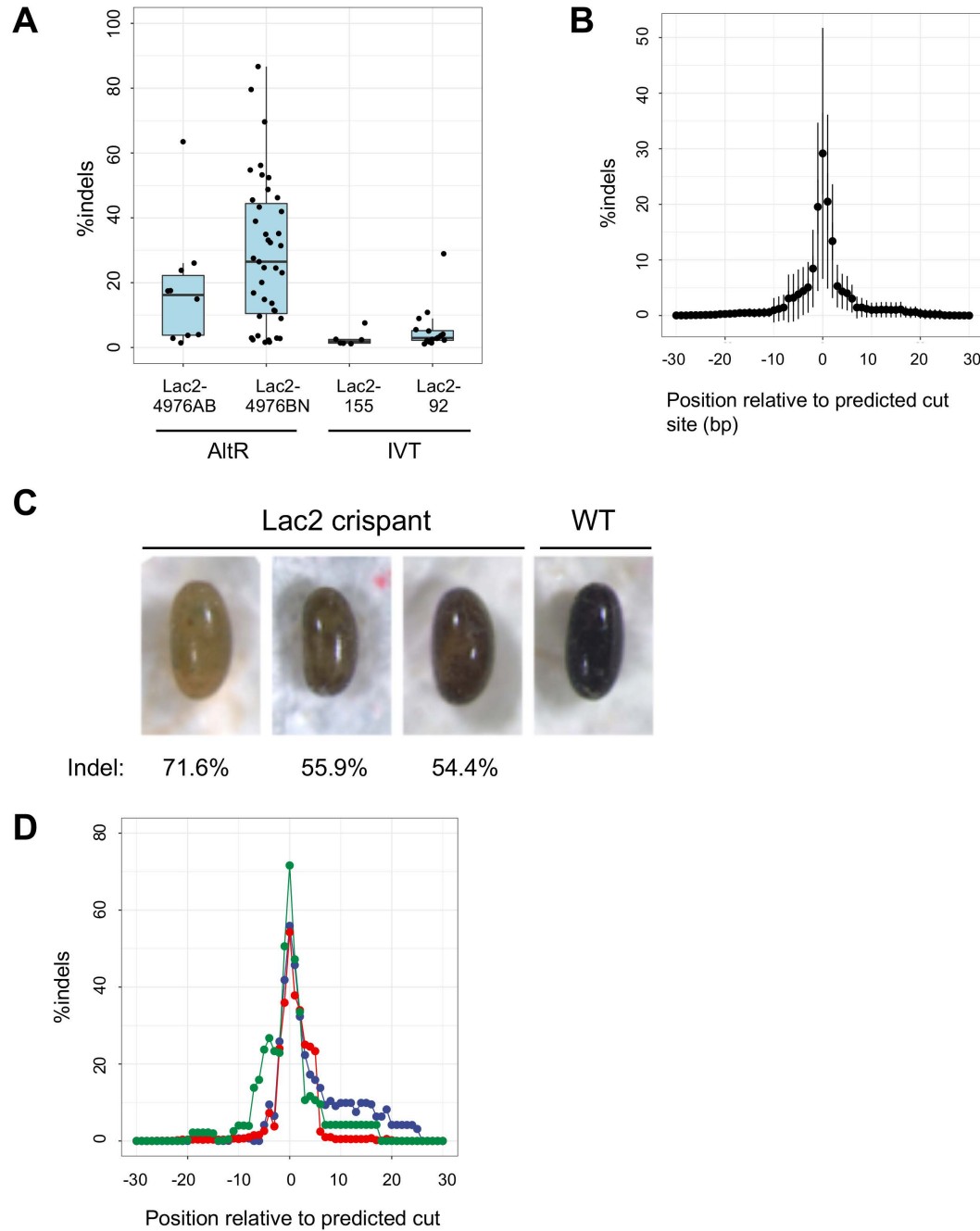

**Fig 4. Targeted disruption of *Lac2* in *A. pisum* (crispant analyses).** (A) The indel percentages of eggs injected with Cas9 and gRNA, as calculated based on amplicon sequencing of individual eggs. See Fig 2A for the target position of each sgRNA. Notably, although 4976BN and Lac2-92 target the same genomic site, 4976BN induced a significant high indel frequency. (B) Quantification of amplicon-seq reads containing indels (insertions and/ or deletions) at the target site. This plot summarizes the data of 39 eggs injected with Lac2-4976BN/Cas9. Data are plotted as means (circles) and standard deviations (lines). (C) Lac2 gRNA (Lac2-4976BN)/Cas9-injected eggs (crispants). Egg photos were captured after diapause (71–77 days after injection). Illumina reads containing indels are indicated at the bottom of each egg image and correspond to the peak values in Fig 2D. (D) Indel distribution profiles from amplicon-seq of the three *Lac2* crispant eggs shown in (C). The percentage of Illumina reads containing indels is plotted relative to the target Cas9 cut site (position 0). Green, blue, and red lines correspond to the first, second, and third egg from the left in panel (C), respectively.

(n = 532) were injected with a mixture of Cas9, crRNA, and tracrRNA; 264 eggs survived. Seventy-one eggs hatched after 2-month embryonic diapause in a cold, moist chamber. Of these, 15 fundatrices grew into adulthood to give birth to the G1 offspring. Each G1 female produced a clonal G2 offspring. We genotyped 45 G1 families by analyzing G2 individuals (genetically identical to G1) using the RGEN-RFLP method, followed by amplicon sequencing. We obtained six stable mutant lines that were classified into unique four lines: 7 bp deletion (mut-1D in Fig 5A), 7 bp deletion (mut-4F), 3 bp deletion (mut-2B), and 5 bp insertion (mut-3E). These mutations were detected at the targeted cleavage site in each line (Fig 5A). All the aphids in these mutant lines carried heterozygous mutations and displayed no apparent lethal defects. The heterozygous lines were maintained in the laboratory as viviparous females that reproduced clonally.

## Phenotypes of heterozygous *Lac2* mutants

All four heterozygous *Lac2* mutants were viable and stably maintained in the laboratory for parthenogenetic viviparous reproduction. Although no noticeable differences were observed between the wild-type and heterozygous mutants, distinct morphological changes were observed in the adult male morphs of the heterozygous *Lac2* mutants. The dorsal surface of the adult male thorax appeared dark in wild-type pea aphids (Fig 6A, left). In contrast, heterozygous *Lac2* mutant males exhibited lighter pigmentation in the dorsal thorax (Fig 6A, right). Additionally, the legs of heterozygous mutant males showed reduced pigmentation compared to those of wild-type males (Fig 6A). Phenotypic differences were also observed between the wings. Wild-type males had straight and flat wings, whereas the wings of heterozygous *Lac2* mutant males were noticeably wrinkled (Fig 6A). The high expression of the *Lac2* transcript in this morph (Fig 3B) may underlie the phenotypic changes observed in male heterozygous mutants.

The established *Lac2* mutant lines were maintained in the laboratory as wingless viviparous females, and no noticeable morphological differences were observed compared with wild-type individuals. However, the color phenotype became evident when we induced a winged viviparous female morph from heterozygous mutant lines under crowded conditions. Similar to the males, heterozygous *Lac2* mutant females exhibited lighter pigmentation in the dorsal thorax than the wild-type (Fig 6B). This reduction in pigmentation was consistent across the independent mutant lines tested (mut-4F, mut-1D, and mut-3E), suggesting that Lac2 contributes to cuticle tanning in a dose-dependent manner. Unlike in males, the legs of heterozygous females did not show a noticeable reduction in pigmentation. Additionally, winged heterozygous mutants induced from the mut-4F or mut-1D lines exhibited malformed wings with a reduced size and wrinkled appearance (Figs 6B and S2A), which was similar to the male phenotype. These findings suggest that *Lac2* is involved in exoskeletal pigmentation and wing formation in winged viviparous female morphs.

## Egg phenotypes of homozygous *Lac2* mutants

**Generation of homozygous** Lac2 **mutants and its unpigmented phenotype.** We cultured aphids under short-day conditions that induced sexual reproduction to generate a homozygous mutant from a stable line (*Lac2* mut-4F) that is heterozygous for a 7-bp deletion in *Lac2* exon 4; males and oviparous females were produced. We inbred the heterozygous male and female mutants to generate homozygous mutants (Fig 5B). The crossed oviparous females produced 119 eggs. Of these, seven eggs were unfertilized and 80 turned black, indicating healthy development. In contrast, 32 eggs exhibited complete loss of melanization (Fig 5B) that is similar but stronger than that of the *Lac2* crispant (Fig 4C). We randomly genotyped 10 eggs (seven black and three unpigmented). All the three unpigmented eggs were homozygous mutants (Fig 5B). These results indicated that the defect in egg pigmentation was caused by the knockout of the *Lac2* locus.

**_Lac2_ mutant eggs failed to hatch while developing up to eye formation stage.** After the successful generation of homozygous *Lac2* mutants from the stable line *Lac2* mut-4F through inbreeding, we generated homozygous mutants from the mut-1D, mut-2B, mut-3E, and mut-4F lines to assess reproducibility across mutant lines (Table 1). Consistent with the initial mut-4 experiments, we obtained two distinct egg color phenotypes: black and unpigmented (Table 1). Genotyping

PLOS Genetics

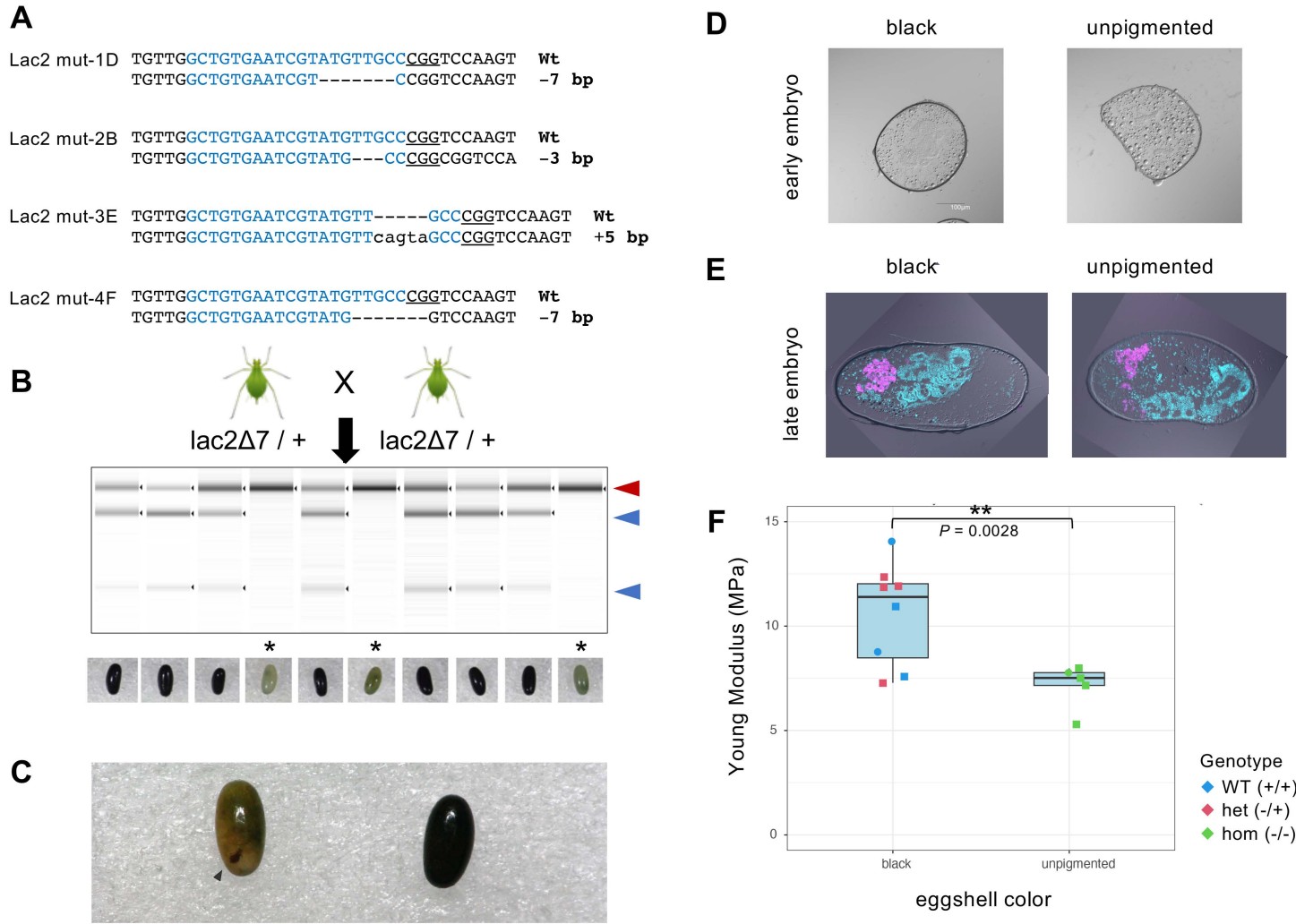

**Fig 5. Germline transmission of *Lac2* mutant alleles, generation of homozygous G1 mutants, and egg phenotypes.** (A) Genotypes of *Lac2* G1 offspring analyzed by amplicon sequencing Targeted guide sequences are highlighted in blue. PAM sequences are underlined. Deletions are indicated by dashes (-) and insertions are indicated by lowercase letters. (B) Generation of a homozygous mutation by self-crossing with a stable line carrying a heterozygous *Lac2* deletion corresponding to mut-4 in (A). The gel image shows the RGEN-RFLP results. Red and blue arrowheads indicate the undigested and digested PCR fragments, respectively. In this experiment, three eggs were found to possess biallelic mutants and are marked with *. Photographs of egg color phenotypes are shown at the bottom. (C) Example of a homozygous *Lac2* mutant egg (left) compared with an uninjected control egg (right). A developed red eye, visible through the eggshell, is indicated by an arrowhead. (D) Differential interference contrast (DIC) images of cross-sections of unpigmented homozygous *Lac2* mutant eggs (right) compared to black eggs (left) prepared from early embryos (12–14 days AEL). Both types of eggs were generated by inbreeding the *Lac2* mut-3E line. (E) Fluorescence in situ hybridization images were overlaid with DIC images of cross-sectioned late embryos (100 d AEL). Both unpigmented and pigmented eggs were generated by inbreeding with the *Lac2* mut-4F line. DAPI signals indicating DNA are colored in cyan, and *Buchnera* symbiont localization was visualized using a Cy5-Apis2a probe colored in magenta. (F) Box plot of Young's modulus (MPa), as measured using atomic force microscopy (AFM). Each point represents an individual egg with the genotype color-coded: homozygous Lac2 mutants (green), heterozygous Lac2 mutants (red), and wild type (blue). Two Lac2 mutant lines (E and F), distinguished by shape, were used in this experiment: squares and circles represent lines E and F, respectibvely. Statistical significance was assessed using one-tailed Welch's t-test, with the p-value (p = 0.0028) indicated between the two boxplots.

confirmed that all unpigmented eggs were homozygous mutants. After genotyping, the remaining eggs were subjected to diapause under winter conditions to monitor hatching. None of the unpigmented eggs hatched across all four mutant lines (80 eggs in total), whereas the black eggs showed a hatching rate of 7.6–47.3% (582 eggs) (Table 1). These results

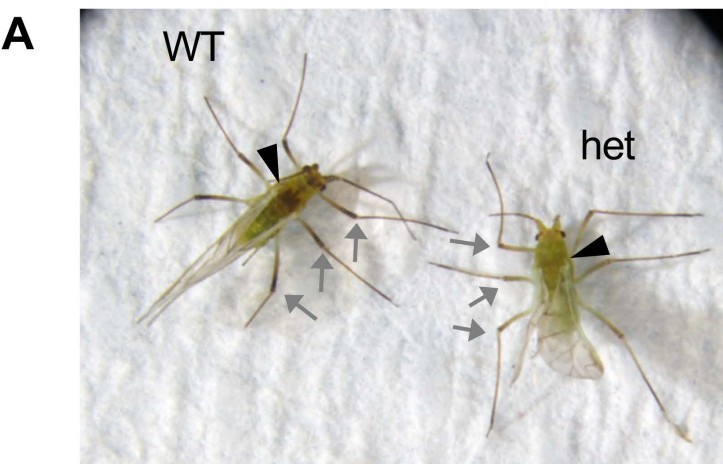

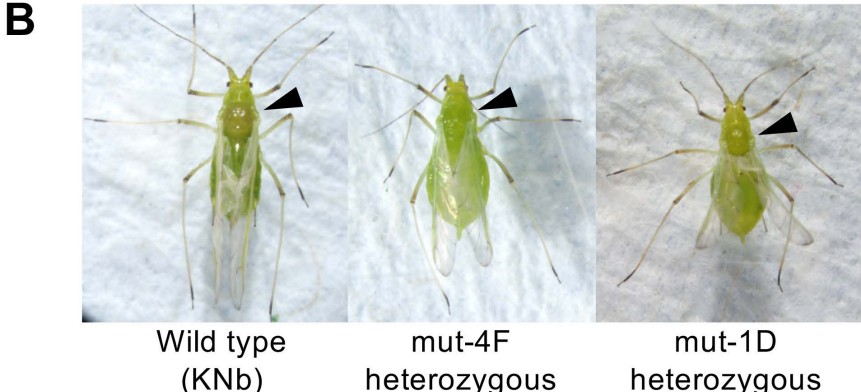

**Fig 6. Phenotypes of heterozygous *Lac2* mutants.** (A) Adult males of a wild type (WT; KNb strain) and heterozygous *Lac2* mutant (mut-4F; labeled as "het"). (B) Winged viviparous females of wild-type (KNb) and heterozygous *Lac2* mutants from the mut-4F and mut-1D lines were induced from wingless morph aphids under crowded conditions. Black arrowheads indicate the thorax, where the tanning level of the cuticle surface is remarkably different between the wild-type and mutants. Grey arrows indicate representative legs showing remarkable different pigmentation levels between the wild-type and the heterozygous mutant.

**Table 1. Egg phenotypes of *Lac2* mutants.**

| Mutant line | Total no. eggs | Black | Un-pigmented | Black (incubated)* | Black hatched | Unpigmented (incubated)* | Unpigmented hatched | Red eye observed |
|---|---|---|---|---|---|---|---|---|
| mut-1D (7-) | 268 | 232 | 36 | 211 | 16 (7.6%) | 30 | 0 (0.0%) | 7 |
| mut-2B (3-) | 342 | 287 | 55 | 256 | 121 (47.3%) | 28 | 0 (0.0%) | 31 |
| mut-3E (5+) | 188 | 167 | 21 | 29 | 29 (21.3%) | 9 | 0 (0.0%) | 2 |
| mut-4F (7-) | 138 | 118 | 20 | 86 | 8 (9.3%) | 13 | 0 (0.0%) | 2 |
| Total | 936 | 804 | 132 | 582 | 174 (29.8%) | 80 | 0 (0.0%) | 42 |

*Black (incubated) and unpigmented hatched (incubated) columns indicate the number of eggs remaining after genotyping (eggs were sacrificed for DNA extraction, followed by PCR or Sanger sequencing) and were subjected to diapause before hatching.

demonstrate that unpigmented eggs are non-viable, indicating that *Lac2* is essential for embryogenesis in overwintering eggs.

Well-developed red eyes were observed through transparent eggshells in 30% of the unpigmented eggs (42 of 132) (Fig 5C and Table 1). Because eye development occurs during the late stages of embryogenesis in oviparous aphids [10], this suggests that *Lac2* is required for developmental processes in the late embryogenesis and/or hatching stages.

**Tanning and sclerotization of eggshells.** Darkening of pea aphid eggs is caused by deposition and tanning of the cuticle by the serosa [11]. Comparison of sections from black and unpigmented eggs revealed that the serosal cuticles of homozygous *Lac2* mutants completely lacked black pigmentation (Fig 5D and 5E).

To quantitatively compare the hardness of black and unpigmented eggshells, we used atomic force microscopy (AFM) to measure Young's modulus, which represents the ratio of stress to strain in a material. A higher Young's modulus indicates greater stiffness and resistance to deformation. The average Young's modulus for unpigmented eggs was 7.14 MPa, whereas that for black eggs was 10.60 MPa. The Young's modulus of unpigmented homozygous *Lac2* mutant eggs was significantly lower than that of black eggs (t = 3.4882, df = 10.266, P = 0.002808) (Fig 5F). Both wild-type and heterozygous *Lac2* mutant eggs exhibited black pigmentation, making it impossible to distinguish between the genotypes during AFM measurements. Therefore, we retrospectively genotyped each egg after AFM analysis. No significant difference was observed in the Young's modulus between the wild-type and heterozygous mutants among the black eggs. These findings suggest that the unpigmented eggshells of homozygous *Lac2* mutants are less stiff than the black eggs, highlighting the critical role of *Lac2* in the hardening of aphid eggshells.

**Egg surface structure and susceptibility to fungal infection.** To investigate the role of *Lac2* in eggshell surface structure, we conducted scanning electron microscopy (SEM) on ten wild-type eggs and five *Lac2*-deficient eggs, all collected within one month after laying (Fig 7A). SEM images of wild-type eggs revealed a smooth and continuous surface with compact and well-organized structural elements, which likely contributed to the protective and waterproofing functions of the eggshell (Fig 7B and 7C). In contrast, 60% of *Lac2*-deficient eggs displayed noticeable dents (Fig 7D), suggesting reduced hardness compared with that of wild-type eggs. Additionally, all *Lac2*-deficient eggs exhibited a distinct phenotype characterized by vein-like raised structures covering the entire surface (Fig 7D and 7E). These raised structures were associated with fungal growth, as conidial chains (bead-like fungal spores) were observed in some *Lac2*-deficient eggs (Fig 7E and 7F). Fungi were not detected in any wild-type eggs. These findings suggest that the absence of *Lac2* compromises the structural integrity and barrier function of the eggshell, making it more susceptible to microbial colonization and fungal infection.

## Development of DIPA-CRISPR for oviparous females in the pea aphid

Recent studies have induced genome editing via NHEJ in fertilized eggs by directly injecting RNPs into the body cavity of several insects, such as cockroaches, red flour beetles, mosquitoes, and whiteflies [25,46,47]. This method, termed DIPA-CRISPR, offers a promising and straightforward alternative to egg injection, and significantly simplifies the injection process. Thus, we further explored the feasibility of DIPA-CRISPR in adult oviparous females of pea aphids.

We injected Cas9 RNPs targeting *Lac2* (sgRNA: Lac2-4976BN) into the body cavities of oviparous female aphids and examined indel mutation rates in the eggs they laid. Nine adult females were randomly selected, injected with Cas9 RNPs, and mated with males. Genomic DNA extracted from eight eggs was analyzed using the Illumina MiSeq platform. Amplicon sequencing revealed indels in 14.6% of the sequences (Fig 8A). As a control, Cas9 RNPs targeting the *BCR3* locus were injected into 14 oviparous females. Among the 64 fertilized eggs obtained, four were randomly selected for amplicon sequencing. *BCR3* is an aphid-specific gene unrelated to *Lac2*, and the corresponding sgRNA has been well characterized. No apparent phenotypic changes were observed in the laid eggs, and amplicon sequencing showed an indel rate close to 0% (0.018%), a level consistent with background PCR or sequencing errors rather than off-target activity [48]. We further evaluated the correlation between egg-laying timing and the induced indel rate; we injected six oviparous females

**A**

| Egg type | Total | Dented | Fungal |
|---|---|---|---|
| Wild-type | 10 | 0 (0%) | 0 (0%) |
| *Lac2* mutant | 5 | 3 (60%) | 5 (100%) |

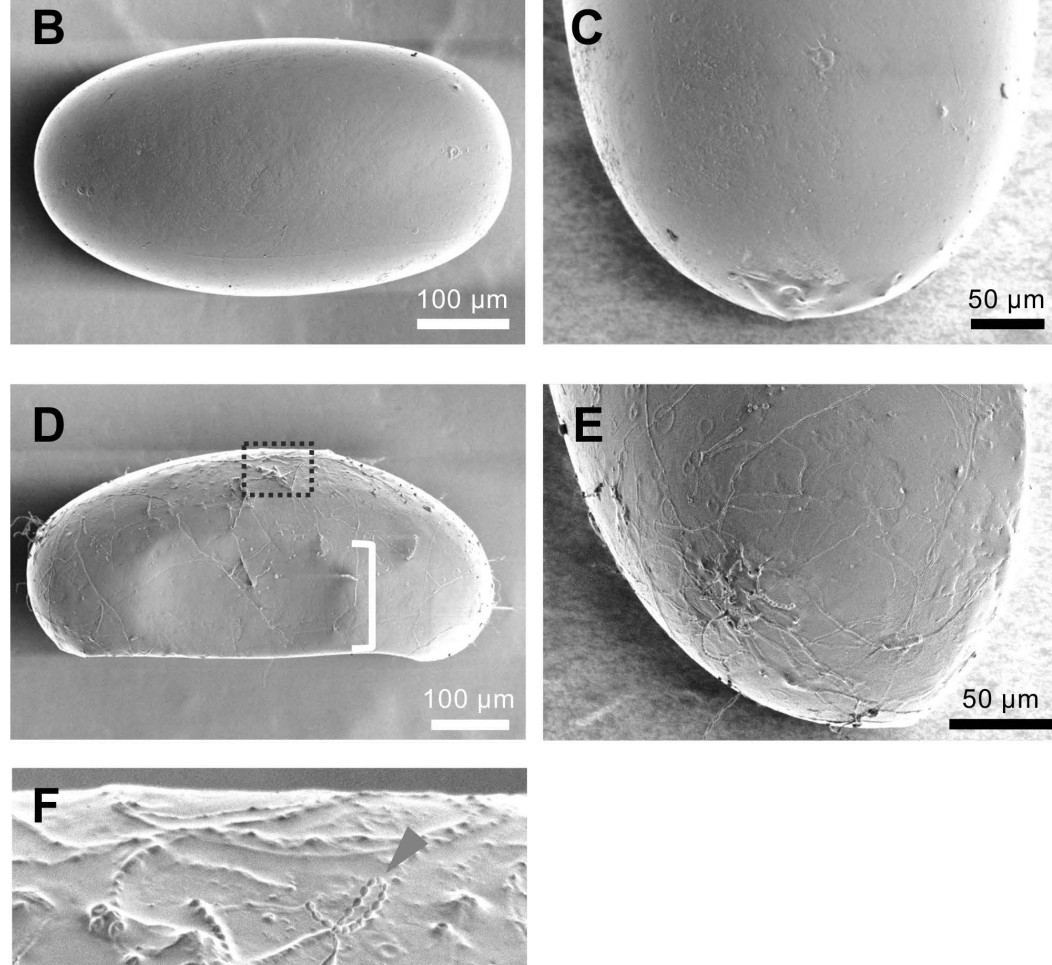

**Fig 7. SEM analysis of the eggshell surface in wild-type and *Lac2*-deficient eggs.** (A) Comparison of fungal growth rates between wild-type and *Lac2*-deficient (Δ7/Δ7) eggs. (B–E) SEM images of the eggshell surface for wild-type (B, C) and *Lac2*-deficient eggs (D, E). Lateral views are shown in (B) and (D), whereas posterior views are shown in (C) and (E). The dented surface areas of *Lac2* mutant eggs are highlighted with a single white outline bracket (D). (F) Magnified view of the area indicated by the dashed line in (D), showing a chain of conidia on the fungal structure.

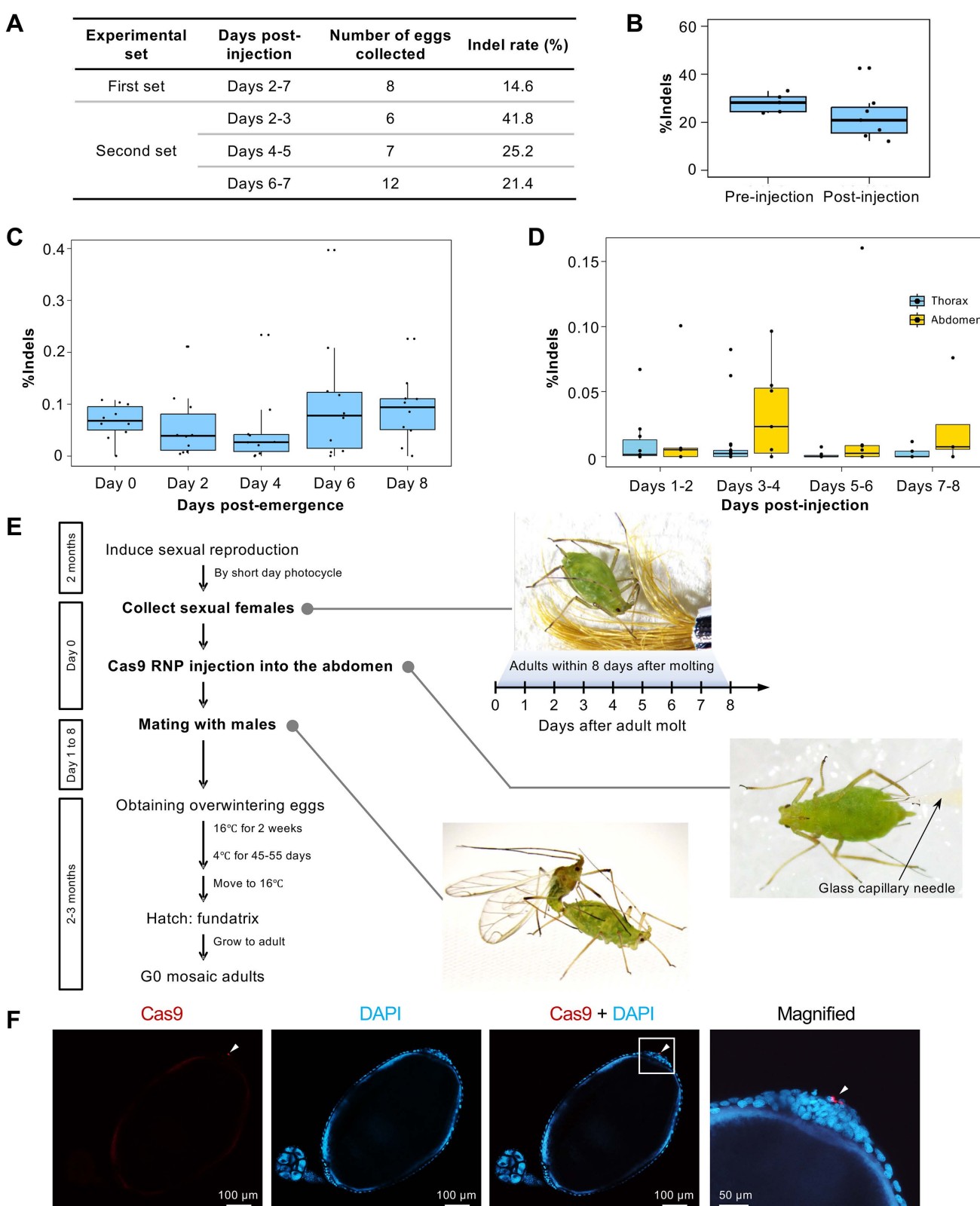

**Fig 8. DIPA-CRISPR for oviparous aphid females and its application in *Lac2* knockout.** (A) Optimization of DIPA-CRISPR: indel rates in fertilized eggs from two DIPA-CRISPR experimental sets of oviparous aphids. The first set shows the indel rates for eggs collected on days 2–7 post-injection.

The second set shows the indel rates for eggs collected at two-day intervals: days 2–3, days 4–5, and days 6–7 post-injection. Genomic DNA from each collection was pooled and analyzed using amplicon sequencing. (B) Comparison of pre- and post-injection RNP: Indel rates in fertilized eggs from pre- and post-injection mated females. (C) Indel rates in fertilized eggs from females injected at different days post-emergence, collected over days 2–7 post injection. (D) Indel rates in fertilized eggs from females injected in the thorax or abdomen, collected at two-day intervals. (E) Established workflow of DIPA-CRISPR for oviparous aphid females. (F) Cas9 RNP uptake: Confocal images showing Cas9 signals at the posterior pole of vitellogenic oocytes, as indicated by a white arrowhead.

and collected eggs at 2-day intervals, with day 1 defined as the day after the injection. The indel rates were 41.8%, 25.2%, and 21.4% for days 2–3 (6 eggs), 4–5 (7 eggs), and 6–7 (12 eggs), respectively (Fig 8A). These results indicate that genome-edited fertilized eggs can be obtained at least one week post-injection.

We investigated the effects of mating timing, optimal timing for Cas9 RNPs injection, and effective injection sites on the body to optimize DIPA-CRISPR in oviparous aphids. We examined the effect of mating on genome editing efficiency by injecting Cas9 RNPs into pre- and post-mating females. Premating females were injected and mated, after which they laid eggs. Amplicon sequencing showed no significant changes in indel rates between pre- and post-mating females, suggesting that mating did not affect the efficiency of mutation induction in fertilized eggs (Fig 8B). We also evaluated the optimal injection timing and target body parts for higher knockout efficiency (S2 Text). Our results indicated that the timing of injection did not significantly affect the knockout efficiency (Fig 8C). However, injections targeting the abdomen resulted in higher indel rates than those targeting the thorax (Fig 8D).

In summary, we successfully established a DIPA-CRISPR method for genome editing in oviparous female pea aphids (Fig 8E). The optimized conditions included injecting Cas9 RNPs into pre-mating females, targeting the abdomen, and administering injections within the first eight days post-adult emergence (Fig 8E).

## Mosaic knockout phenotype of DIPA-CRISPR-mediated *Lac2* mutation

We obtained two mosaic *Lac2* mutant eggs from Cas9 RNP-injected mothers using the newly established DIPA-CRISPR method for oviparous aphid females (Fig 9A). Both eggs exhibited a mosaic pattern of black pigmentation loss. Because *Lac2* is the dominant gene in pigmentation, the absence of black pigmentation in these regions indicates a biallelic KO, suggesting that both *Lac2* alleles were disrupted in these areas. The indel rates in the two eggs were 94.2% and 60.4%, respectively. These were associated with a marked reduction in black pigmentation on the egg surface (Fig 9A). These results suggest that biallelic KO mutations were induced early during embryogenesis rather than during oogenesis.

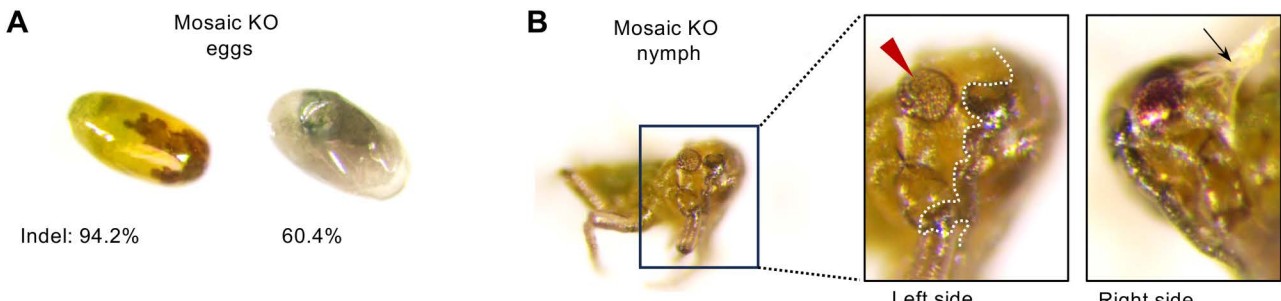

**Fig 9. Mosaic *Lac2* mutant phenotypes obtained through DIPA-CRISPR.** (A) Mosaic analysis of *Lac2* knockout fertilized eggs. Indel rate and phenotypic analysis of two mosaic eggs with indel rates of 94.2% and 60.4%, respectively. Significant inhibition of black pigmentation is shown. (B) G0 mosaic nymphs (first-instar stem mothers) with *Lac2* knockout. The nymph exhibited inhibited pigmentation in the cuticle, specifically in the area to the left of the white dotted line and in the compound eyes indicated by a red arrowhead. Additionally, the nymphs exhibited abnormal locomotion owing to incomplete embryonic molting. The black arrow indicates an attached old cuticle due to incomplete embryo molting.

One egg laid by a mother treated with DIPA-CRISPR targeting the *Lac2* locus successfully hatched after diapause, yielding a G0 mosaic mutant fundatrix nymph. However, this G0 nymph died shortly after hatching and did not undergo the first molt. The G0 nymphs exhibited mosaic inhibition of pigmentation in both the cuticle and compound eyes, where red pigmentation was disrupted (Fig 9B). The G0 mutant nymph also displayed a thin membrane that partially adhered to the body surface (Fig 9B) and exhibited abnormal locomotion (S1 Movie). Given the known role of *Lac2* in cuticle formation and hardening in other insects, it is likely that the knockout led to defective cuticle formation, resulting in incomplete embryonic molting in G0 nymph. The relatively high expression of *Lac2* in late diapause eggs of (1.5–2 months AEL) (Fig 3A) may account for the defects observed in embryonic molting in the *Lac2* knockout mutants generated via DIPA-CRISPR.

## Cas9 RNP incorporation into oocytes in DIPA-CRISPR

To examine how injected Cas9 RNPs are incorporated into developing oocytes, we performed immunohistochemistry using an anti-His tag monoclonal antibody to detect Cas9 localization. Because Cas9 was injected as a part of the Cas9 RNP complex, the detected Cas9 signals indicate Cas9 RNP localization. Ovarioles were collected 24 h after the Cas9 RNP injection into oviparous females. Our immunohistochemical analysis revealed a distinct localization of Cas9 RNPs within the developing aphid oocytes. Cas9 signals were predominantly detected in large spherical follicular cells at the posterior poles of vitellogenic oocytes. These posterior signals were consistent and clear, suggesting a robust uptake mechanism at this site (Fig 8F).

Z-stack imaging revealed that the signal at the posterior pole comprised multiple signals (S3A Fig), suggesting that they were clustered. This clustering of Cas9 RNPs implies that they may bind to specific cellular structures or protein complexes, providing insights into the mechanisms underlying RNP uptake and intracellular transport pathways. Additionally, Cas9 signals were detected along the lateral regions of the oocytes in follicular cells (S3B Fig). Although these lateral signals were less distinct and varied in intensity, they indicated that Cas9 RNPs may be taken up through pathways other than those at the posterior pole. The presence of these signals suggested the possibility of multiple entry points or mechanisms for Cas9 RNP incorporation into oocytes.

## Discussion

This study investigated the roles of the Laccase2 (*Lac2*) in the pea aphid (*A. pisum*) using CRISPR/Cas9-mediated gene knockout. Our optimized genome editing protocol allowed us to successfully generate *Lac2* mutants, revealing multiple phenotypic effects. These include deficiencies in melanization, eggshell hardening, and protection against fungal infection in overwintering eggs as well as reduced pigmentation in the thoracic cuticle and impaired wing development in winged adult morphs. In the following sections, we will discuss the functions of the *Lac2* gene and the development of CRISPR/Cas9 genome editing in the pea aphid in greater detail.

### Functions of *Lac2* in the pea aphid

Our CRISPR/Cas9-mediated disruption of *Lac2* revealed its essential roles in multiple developmental stages and morphs of the pea aphid. Homozygous knockouts resulted in pigment-less, soft-shelled eggs that were embryonically lethal (Fig 5C–5F), while G0 and heterozygous mutants exhibited phenotypes such as reduced cuticle pigmentation, malformed wings, and molting defects (Figs 4C,6,9A and 9B). These findings demonstrated the multiple critical roles of *Lac2* in different developmental stages and morphs of the pea aphid.

### Essential roles of *Lac2* in overwintering eggs and environmental adaptation

The Lac2 enzyme plays a crucial role in sclerotization and pigmentation of insect cuticles [2,6,39,44,49–51]. Tanning caused by *Lac2* is also essential for other structures such as eggshells, chorions, oothecae, and butterfly pupa [52–54]. In this study, eggs injected with *Lac2* gRNA/Cas9 showed reduced pigmentation (Fig 4C), and loss-of-function *Lac2* mutants

generated by crossing stable heterozygous *Lac2* mutants exhibited a complete lack of eggshell pigmentation (Fig 5B–5E), with none of the eggs hatching (Table 1). The expression pattern of *Lac2* mRNA in aphid eggs coincided with the onset of eggshell pigmentation (Fig 3A), indicating that *Lac2* is required for the pigmentation of pea aphid eggs. In addition, the unpigmented eggshells of homozygous *Lac2* mutants were significantly less stiff than pigmented eggs (Fig 5F), highlighting the critical role of *Lac2* in the hardening of aphid eggshells. Similar phenotypes have been reported in other insects such as the mosquito (*Aedes albopictus*), in which *Lac2* is essential for eggshell sclerotization and pigmentation [54]. The colorless eggshell phenotype in mosquitoes closely resembles that observed in pea aphids in this study.

The failure of unpigmented aphid eggs to hatch, and their consequent death (Table 1), indicate the essential role of *Lac2* in embryonic development in pea aphids. The visibility of red eyes through colorless eggshells suggests that the embryos developed to a late stage, as eye formation occurs during the final stages of aphid embryogenesis [10]. Cryo-sectioning of late-stage diapause eggs confirmed the presence of developed embryos (Fig 5E). However, the embryos did not hatch. The exact cause of mortality remains unclear. One possibility is that *Lac2* knockout embryos lack sufficiently hardened cuticles for successful hatching, owing to defects in sclerotization. Alternatively, the loss of eggshell protective function may lead to accumulated damage during prolonged diapause. The damage can be a physical, fungal, or bacterial pathogenic infection. Indeed, we observed a higher incidence of fungal infection on the surface of *Lac2* mutant eggs than in wild-type eggs (Fig 7). These observations indicate that *Lac2* plays essential roles in the survival and development of overwintering eggs through five key functions: eggshell pigmentation, eggshell hardening, late-stage embryonic development, hatching, and protection against fungal infections. These functions represent ecological and evolutionary adaptations in the complex life cycle of pea aphids, including overwintering diapause and cyclic parthenogenesis.

### *Lac2* roles against fungal infection

Pathogenic fungi pose a major threat to overwintering aphid survival. In the bird cherry-oat aphid *(Rhopalosiphum padi)*, approximately 60% of overwintering eggs in the wild die, primarily owing to fungal infections [55]. Similarly, fungal infections rapidly kill eggs without ant grooming in hinoki cypress-feeding aphids (*Stomaphis hirukawai*) [56]. These examples show the importance of strategies that protect aphid eggs from fungi, whether through mutualistic relationships with ants or intrinsic defenses within the eggshell. Our study demonstrated that *Lac2* inhibits fungal development in pea aphid eggshells (Fig 7). Extensive fungal growth was observed on the surface of aphid eggs in the absence of *Lac2*, suggesting that the enzyme is essential for maintaining eggshell integrity and preventing fungal colonization.

Lac2 catalyzes the oxidation of phenolic compounds, consequently producing quinones that facilitate the crosslinking of chitin and cuticular proteins [4]. This cross-linking is crucial for creating a hardened cohesive cuticle that acts as a barrier against environmental factors such as moisture and microbial invasion [57,58]. This crosslinking was likely disrupted in *Lac2*-deficient eggs, leading to vein-like fungal growth patterns (Fig 7D–7F). These fungal structures suggest that the weakened and uneven eggshell surfaces provide sites for fungal spores to adhere and proliferate.

Although our study showed that *Lac2* is critical for eggshell integrity and fungal resistance, SEM analysis revealed no obvious surface irregularities in *Lac2*-deficient eggs at the magnifications used. Thus, higher magnification and resolution may be required to detect microstructural changes. Surface depressions or cracks can create niches for fungal spores and promote fungal growth [57,59]. Therefore, future studies using high-resolution imaging are required to further elucidate the microstructural impact of *Lac2* deficiency.

**Conserved role of** Lac2 **among insects: cuticle sclerotization, pigmentation, and wing formation.** Our findings demonstrate that *Lac2* is critical for proper cuticle tanning and wing formation in pea aphids. Heterozygous *Lac2* mutants exhibited significant morphological changes, particularly in adult males, who displayed lighter pigmentation on the dorsal thorax and legs, along with wrinkled wings (Fig 6A). Similar phenotypes were observed in the winged viviparous females (Fig 6B). These observations are consistent with phenotypic changes observed in other insects. For example, *Lac2* loss results in defective cuticle hardening and wing abnormalities in red flour beetle [2]. Similarly, *Lac2* knockdown leads to

soft, unpigmented cuticles. This demonstrates the broadly conserved role of *Lac2* in cuticle development across insect species [44,50].

The dose-dependent effects of *Lac2* in heterozygous mutants, which are viable yet display reduced pigmentation, demonstrate the importance of precise *Lac2* regulation for normal development. The complete loss of *Lac2* has been associated with more severe defects and lethality in other insect models, such as *A. albopictus*, where *Lac2* knockdown resulted in soft, unpigmented cuticles and impaired eclosion [51]. Similarly, *Lac2* dosage influenced the degree of tanning and overall cuticle strength in *T. castaneum* [2].

### CRISPR-Cas9-mediated genome editing in the pea aphid

**Optimization of CRISPR-Cas9 mediated genome editing protocol for unique aphid biology.** This study presents an efficient approach for targeted mutagenesis and generation of biallelic (homozygous) knockouts in *A. pisum*. We used CRISPR/Cas9 to induce mutations in the pea aphid genome via the NHEJ pathway. Our optimized protocol (Fig 1) achieved highly efficient gene disruption even in the founder generation (Fig 4). Amplicon-seq analysis of the on-target sites revealed somatic mutation rates as high as 80% in most crispants, allowing for direct phenotypic analysis (Fig 4C). We also successfully generated stable mutant lines through germline transmission of the mutations (Fig 5A). Furthermore, we generated homozygous mutants by inbreeding stable heterozygous lines (Fig 5B), which requires another cycle of sexual reproduction and several months. Therefore, both high-mutation crispants and stable mutant lines could be used within a reasonable timeframe for functional gene analysis in pea aphids (Fig 1).

Despite the success of CRISPR/Cas9 in various insect species, the unique biology of aphids presents challenges for genome editing. These challenges include: (1) a complex life cycle involving both asexual viviparous and sexual oviparous reproduction, (2) an obligatory diapause period, (3) symbiosis with endosymbiotic bacteria, (4) pronounced inbreeding depression, and (5) high nuclease activity. Le Trionnaire et al. [23] addressed these issues in their pioneering study, which marked the first successful application of CRISPR/Cas9 in pea aphids, as exemplified by genome editing of *stylin-01*, a cuticular protein gene. Although our protocol and that of Le Trionnaire et al. (hereafter referred to as LT2019) were developed independently [23], they share several similar strategies.

Both protocols configure long- and short-day incubators to optimize sexual morph induction and maximize egg collection to manage the complex life cycle of aphids. This involves alternating sexual and asexual reproduction in response to seasonal photoperiod changes. The efficient induction of sexual morphs is critical for producing sufficient eggs for CRISPR/Cas9 injection. Both protocols employ outcrossing strategies with different strains to mitigate inbreeding depression [26]. Le Trionnaire et al. [23] used five strains from eastern France to identify the best outcrossing combinations. Similarly, we tested nine strains from Sapporo, Japan, and identified the most compatible cross between the ApL and KNb (or 09003A) strains. Outcrossing is essential for adequate hatching rates and fertility, as it mitigates the severe inbreeding depression observed in pea aphids [26].

Although the overall workflows of both protocols were similar, there were some distinct differences. For diapause management, LT2019 maintained eggs on *Vicia fabae* leaves transferred from glass slides on which injection was performed, whereas we used moist chambers with double-sided adhesive tape to hold the eggs after microinjection. This approach reduces labor and minimizes the risk of egg damage through manipulation. For symbiont preservation, LT2019 injected CRISPR components into the anterior part of the egg to avoid damage to the symbionts at the posterior pole. In contrast, we injected areas adjacent to the symbionts, near the posterior region, to enhance germline cell exposure while minimizing damage. Additionally, although LT2019 secures 90-day diapause period (5 days at 15°C and 85 days at 4°C), which is comparable to natural conditions, we shortened the diapause period to 59 days (2 weeks at 16°C and 45 days at 4°C) based on a previous study [10]. This shortening of the workflow accelerates mutant generation and reduces the risk of egg damage from desiccation or fungal infections during prolonged incubation. Moreover, our approach for mitigating high nuclease activity is unique. We used the Alt-R tracrRNA system from IDT, which significantly improved mutation

efficiency compared with *in vitro*-transcribed gRNAs (Fig 4A). The Alt-R system incorporates 2′-O-methyl and phospho-rothioate modifications to tracrRNA ends and proprietary chemical modifications to crRNA, thereby increasing resistance to nucleases and reducing immune responses. This resistance likely enhances gRNA stability and function because the RNase-rich environment of aphids rapidly degrades dsRNA [45]. In summary, although minor differences were observed between our protocol and that of LT2019, the key to successful genome editing in the pea aphid is in effectively managing its complex life history and addressing its unique biological traits.

Although this study focused on the knockout of *Lac2*, our protocol is sufficiently robust and efficient for application to other genes. To date, we have successfully knocked out ten different genes using this method with specifically designed crRNAs, yielding approximately 100 germline mutants. Our protocol for pea aphids could also be extended to other aphid species, if sufficient eggs can be obtained in the laboratory or in the field. However, further optimization may be required for each species.

**DIPA-CRISPR development in oviparous pea aphids.** This study also developed a DIPA-CRISPR protocol by injecting CRISPR-Cas9 RNPs into oviparous adult aphids (Fig 8E). To our knowledge, this is the first successful application of DIPA-CRISPR in aphids. DIPA-CRISPR is markedly simpler than egg microinjection, as it eliminates labor-intensive egg collection at specific developmental stages, along with the specialized skills required for microinjecting small eggs. This method enabled mosaic analysis in the G0 generation, providing a powerful tool for studying lethal homozygous genes, such as *Lac2* (Fig 9). Although the current DIPA-CRISPR protocol shows a lower editing efficiency than egg-based methods and has not yet achieved germline-inherited mutations, it offers a practical alternative for genome editing in pea aphids by reducing time and technical demands.

In aphids, asexual reproduction takes place through viviparity, meaning genetically identical offspring develop inside the mother and are delivered live instead of being laid as eggs. Therefore, conventional genome editing approaches that target externally laid eggs cannot be applied directly to the asexual generation. Although genome editing of asexual individuals would offer significant advantages, such as eliminating the need for diapause and avoiding issues related to inbreeding, its feasibility remains a challenge. Future work may explore the possibility of adapting DIPA-CRISPR for use in viviparous asexual females, potentially expanding the applicability of gene editing in aphid research.

Although our optimization efforts focused on mating timing, injection timing, and injection site (Fig 8A–8D), further refinement was required to enhance the mutation rates. One potential improvement could be an increase in the concentration of Cas9 RNPs, as higher doses may improve gene-editing efficiency in DIPA-CRISPR [25]. Another potential strategy involves the use of older, oviparous females. Oocyte maturation in older females may differ from that in younger females, potentially influencing the uptake and effectiveness of Cas9 RNPs. Although oocyte development is asynchronous [60], the relationship between aging and oocyte maturation in aphids remains largely understudied. Studies on other hemipterans, such as the southern green stink bug (*Nezara viridula*), have shown that ovarian development and physiological changes occur gradually and are age-dependent [61]. In addition, vitellogenic oocyte patency, in which follicle cells create gaps that allow the yolk and other macromolecules to enter the oocyte [62], is crucial for Cas9 RNP uptake during oocyte development [25]. Because older oviparous females typically contain more vitellogenic oocytes [63], their use could lead to more effective Cas9 RNP incorporation and an increased number of successfully genome-edited eggs.

Immunohistochemical analysis revealed significant Cas9 signals in follicular cells at the posterior pole of vitellogenic oocytes (Fig 8F). The posterior pole of the oocyte exhibits patency during late vitellogenesis, consequently facilitating the incorporation of the endosymbiont *Buchnera*. This patency is characterized by the spaces between the posterior follicular cells and the perivitelline separating the follicular cells from the oocyte [64]. Egg envelopes composed of the vitelline membrane and chorion form only in the apical and lateral regions of the oocyte in pea aphids, leaving the posterior region exposed [60]. In the rice planthopper *Nephotettix cincticeps*, vitellogenin precursors enter the oocyte in a receptor-independent manner. They bind to the entry path of the endosymbiont *Nasuia*, presumably by interacting with a surface channel molecule on *Nasuia* [65]. This example illustrates how different molecules can utilize endosymbiont pathways to

enter oocytes. Clustering of the Cas9 signals at the posterior pole in oviparous aphids suggested a similar mechanism involving *Buchnera* (Fig 8F). However, further studies are required to explore the role of patency in endosymbiont uptake in oviparous aphids, as this could provide insights into improving Cas9 RNP incorporation and enhancing DIPA-CRISPR efficiency in this insect.

## Supporting information

**S1 Text. IVT gRNA-mediated CRISPR/Cas9 genome editing.**
(PDF)

**S2 Text. Optimization of DIPA-CRISPR in oviparous aphids.**
(PDF)

**S1 Fig. Phylogenetic tree of laccase and multicopper oxidase (MCO) sequences.**
(PDF)

**S2 Fig. Phenotype of heterozygous Lac2 mutants.**
(PDF)

**S3 Fig. Localization of Cas9 RNPs in oviparous aphid oocytes.**
(PDF)

**S1 Table. Lac2 homologs used in phylogenetic analysis.**
(XLSX)

**S1 Movie. Abnormal locomotion in a G0 mosaic first-instar nymph.**
(MP4)

## Acknowledgments

We thank Dr. David L. Stern (HHMI Janelia Research Campus) for valuable support in developing the CRISPR-Cas9 genome editing technique and Dr. Shin-ichi Akimoto (Hokkaido University) for providing the aphid strains. We thank Drs. Takaaki Daimon and Yu Shirai (Kyoto University) for their crucial support in developing the DIPA-CRISPR technique. We thank the NGS team at the National Institute for Basic Biology (NIBB) Tran-Omics Facility for NGS sequencing. We also thank Drs. Teruyuki Niimi, Shinichi Morita and Ken-ichi Suzuki (NIBB) for their advice on genome editing and for sharing research instruments, including SEM. We also thank Drs. Kota Ogawa (Kyushu University) and Aishwarya Korgaonkar (HHMI Janelia) for their valuable discussions.

## Author contributions

**Conceptualization:** Shuji Shigenobu, Shinichi Yoda.

**Data curation:** Shuji Shigenobu, Shinichi Yoda.

**Formal analysis:** Shuji Shigenobu, Shinichi Yoda, Sonoko Ohsawa, Miyuzu Suzuki.

**Funding acquisition:** Shuji Shigenobu, Shinichi Yoda.

**Investigation:** Shuji Shigenobu, Shinichi Yoda, Sonoko Ohsawa, Miyuzu Suzuki.

**Methodology:** Shuji Shigenobu, Shinichi Yoda, Sonoko Ohsawa, Miyuzu Suzuki.

**Project administration:** Shuji Shigenobu.

**Resources:** Shuji Shigenobu.

**Software:** Shuji Shigenobu.

**Supervision:** Shuji Shigenobu.

**Validation:** Shuji Shigenobu, Shinichi Yoda, Miyuzu Suzuki.

**Visualization:** Shuji Shigenobu, Shinichi Yoda, Miyuzu Suzuki.

**Writing – original draft:** Shuji Shigenobu, Shinichi Yoda, Sonoko Ohsawa, Miyuzu Suzuki.

**Writing – review & editing:** Shuji Shigenobu, Shinichi Yoda, Miyuzu Suzuki.

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
