## [Decision Letter · Decision Letter 0]

PGENETICS-D-24-01533

Refined CRISPR/Cas9 genome editing in pea aphids uncovers the essential roles of Laccase2 in overwintering egg adaptation

PLOS Genetics

Dear Dr. Shigenobu,

Thank you for submitting your manuscript to PLOS Genetics. After careful consideration, we feel that it has merit but does not fully meet PLOS Genetics's publication criteria as it currently stands. Therefore, we invite you to submit a revised version of the manuscript that addresses the points raised during the review process.

Please submit your revised manuscript within 60 days Apr 19 2025 11:59PM. If you will need more time than this to complete your revisions, please reply to this message or contact the journal office at plosgenetics@plos.org. Please include the following items when submitting your revised manuscript:

We look forward to receiving your revised manuscript.

Kind regards,

Subba Reddy Palli, Ph.D.

Academic Editor

PLOS Genetics

Quanjiang Ji

Section Editor

PLOS Genetics

Aimée Dudley

Editor-in-Chief

PLOS Genetics

Anne Goriely

Editor-in-Chief

PLOS Genetics

**Journal Requirements:**

At this stage, the following Authors/Authors require contributions: Shuji Shigenobu, Shinichi Yoda, Sonoko Ohsawa, and Miyuzu Suzuki. Please ensure that the full contributions of each author are acknowledged in the "Add/Edit/Remove Authors" section of our submission form.

The list of CRediT author contributions may be found here: https://journals.plos.org/plosgenetics/s/authorship#loc-author-contributions

- ® on page: 16

- TM on page: 16.

4) Please upload a copy of Figure 2d which you refer to in your text on page 28. Or, if the figure is no longer to be included as part of the submission please remove all reference to it within the text.

Potential Copyright Issues:

i) Please confirm (a) that you are the photographer of 3a, 4c, 5b, 5c, 6, 8c, 9, and S2, or (b) provide written permission from the photographer to publish the photo(s) under our CC BY 4.0 license.

ii) Figures 2b, and 5b. Please confirm whether you drew the images / clip-art within the figure panels by hand. If you did not draw the images, please provide (a) a link to the source of the images or icons and their license / terms of use; or (b) written permission from the copyright holder to publish the images or icons under our CC BY 4.0 license. Alternatively, you may replace the images with open source alternatives. See these open source resources you may use to replace images / clip-art:

6) Please ensure that the funders and grant numbers match between the Financial Disclosure field and the Funding Information tab in your submission form. Note that the funders must be provided in the same order in both places as well. Currently, the order of the funders is different in both places.

**Reviewers' comments:**

Reviewer's Responses to Questions

Reviewer #1: This paper established an efficient genome editing system using CRISPR/Cas9 in the pea aphid targeting the Laccase2 gene. Since it has been challenging to edit the genome of aphids due to their complex life cycles, this system will provide useful information. It also provides important insights into the role of Laccase2 in aphids, especially on egg pigmentation, eggshell hardening and wing formation.

Overall, the experiments are well designed and carefully analyzed. On the other hand, the description is complicated due to the combination of several experiments, and some information is missing, making it difficult to follow. Also, the name of Laccase should be revised as it may cause confusion.

The following comments are provided mainly concerning the description of the manuscript.

1.

Laccase was originally distinguished into two groups, laccase1 and laccase2, but since laccase1 has no laccase activity, this name is no longer used and it is more appropriate to refer to it as MCO (see Dittmer & Kanost, 2010). The name Laccase2 can be left as it is, but please revise the description in the text as Laccase1 should be described as MCO1 in Fig. 2, Fig. S1, etc. Also, would the phylogenetic tree include MCORPs involved in ovarian development in Tribolium castaneum (Peng et al, 2014). Authors should also specify the relationship to MCORP when constructing the molecular phylogenetic tree.

2. The author addresses the issue of Complex life cycle and Inbreeding depression on Page 27, but please describe what specific conditions were used. It is unclear how the authors set the day length and temperature for the summer and winter conditions. Also, the authors noted that they combined eight aphid lines, please specify the actual results and which combination was selected as the better one.

3. Please specify if the sgRNA for IVT was designed in the same position as for Alt-R. If it was designed in a different position, this may also affect the results. Is the crRNA used for DIPA-CRISPR the same Lac2-4976BN used for the egg injection? Please specify how it was prepared.

Other minor comments are as follows.

>Title

pea aphids -> the pea aphid

>Page 8, line 175

Please describe the full name of AEL.

>Page 19, line 439

Is “The statistical” a typo?

>Page 29, line 662

What exactly is the “appropriate humidity level”?

>Page 32, line 758

Please add arrows to the legs in Fig. 6A.

>Page 33, line 761

Fig 3A -> Fig 3B

>Page 33, line 780

appendages -> legs (?)

>Fig 1

Please describe the conditions of the day length in detail.

>Fig 2A

The authors have designed two crRNAs, please specify which one.

>Fig 5C

Please add arrows to indicate red eyes.

>Fig 8 legend

What does (Yoda) mean?

>S3 Fig

This is important information and should be moved to the main figure.

Reviewer #2: Shigenobu et al. reported genome editing techniques in the pea aphid, which is a model organism in evolution, ecology, development, and agriculture. They developed a knockout technique using microinjection into fertilized eggs, enabling more stable, efficient, and rapid mutant generation compared to conventional methods. Additionally, for the first time in aphids, they introduced the simpler DIPA-CRISPR method, which is easier than microinjection into eggs. Methods for 1) preparation of eggs for microinjection, 2) introduction of the exogenous materials into eggs or bodies and 3) genome editing must be developed in a species- and strain-specific manner. The authors have made very careful observations, preparations, and evaluations. These advancements in genetic manipulation technologies are expected to significantly enhance the utility of this aphid as a model organism.

A previous study by Le Trionnaire et al. on genome editing in aphids has already been published. The authors cited this paper and discussed the similarities, differences, and advantages of their method in the Discussion section. They rigorously evaluated the genotypic and phenotypic changes resulting from introducing mutations in the target lac2 gene, providing strong evidence to support their claim that this method is highly effective.

The primary concern of this reviewer is that despite being submitted as a Method paper, the study seems to place greater emphasis on the functional analysis of the lac2 gene in overwintering eggs. While this is the first functional analysis of lac2 in overwintering eggs, the role of lac2 in the pigmentation of eggs has already been reported in other insects, such as mosquitoes. This raises questions about the originality of the finding of the lac2 function in the aphid. Therefore, this reviewer suggests that the authors focus more on the development of genome editing methods. Additionally, this reviewer recommends that the authors highlight the limitations of previous aphid genome editing techniques in the Introduction and describe how this study addresses and improves upon these issues.

Minor comments/suggestions

Line 399: Why did the authors choose the BCR3 gene as a control for Cas9 RNPs targeting? Please explain the reason and the gene’s function.

Line 439: “The statistical” may be omitted.

Line 530: It would be better to mention why the presence of the signal peptide is important for the Lac2 function.

Lines 675-677: The authors performed genotyping of the survived eggs. Please explain which stage of embryos were used for genotyping.

Lines 708-709: How was the amount and ratio of Cas9, crRNA, and tracrRNA determined?

Lines 710-717: Six stable mutant lines were obtained from 15 Fundatrix. Please explain how many Fundatrix produced the mutant lines.

Lines 755-760: mut-4F male showed reduced pigmentation. mut-4F and mut-1D showed winkled wings. Please explain the phenotypes of the other mutants that were not described in the manuscript. If the non-frameshift mutation in the mu-2B did not change the phenotypes in males, the importance of the frameshift mutation could be emphasized.

Lines 885-886: How many control animals injected with BCL2 crRNA were prepared? What was the result of the control experiment?

Lines 894-896: The indel rates of eggs in the DIPA-CRISPR experiments were shown here. Were these eggs collected from the same mother or several different mothers? Because the authors injected Cas9 RNPs into nine adult females, information on the difference in the efficiency of DIPA-CRISPR between individuals helps to understand the reproducibility of this method.

Lines 932-933: The two mosaic Lac2 mutants were from the same mother or two different mothers?

Line 1118: It would be nice for readers to know why asexual eggs are not useful for genome editing in the aphid.

Fig 4D: There is no description of the difference in color of each line.

Fig 8 legend: (Yoda) may not be necessary.

Reviewer #3: This paper reports (1) improved protocols using CRISPR/Cas9 with aphids and (2) an experimental analysis of the role of the laccase 2 gene in aphid biology.

Aphids are a prime model for developmental plasticity, in which the same genotype can produce a variety of distinct female morphs based on environmental cues during early development. Aphids are also a central model for heritable symbiosis, with some hundreds of papers on the aphid/Buchnera system. Finally, they are major crop pests and are used in studies of host plant recognition and specialization and in studies of speciation mediated by host plant shifts. The pea aphid has come to be the primary species used for studies on this group and is studied by labs around the world. It has a chromosome-level genome RefSeq assembly.

Despite their importance, there are relatively few experimental genetics studies on aphids due to the unique challenges of their life cycles, as detailed in this manuscript. RNAi has had only limited success due to the high RNase activity in aphid tissues. Germline modification is challenging as the asexual phase does not provide eggs that can be injected, and the sexual phase involves induction of sexual morphs (~2 months), mating them, then a long egg diapause (~ 3 months).

The protocols described here for CRISPR/Cas9 will be enormously useful. The manuscript describes everything a researcher would need to know for knocking out target genes in pea aphid, from how to continuously produce the sexual forms and achieve good levels of egg hatching to the details of designing and generating the CRISPR constructs and injection techniques themselves. The presentation of both an egg injection protocol and the DIPA-CRISPR protocol for injecting sexual mothers to obtain mutant progeny.

The findings on the Lac2 functions are well-supported and interesting, even if not very surprising. This was a good choice for demonstrating the efficacy of these methods, as it enabled visualization of the various outcomes such as the mosaic embryos of the DIPA-CRISPR methods (shown in Fig. 9).

I have no major suggestions for changing the manuscript. I have listed below some minor issues.

Minor comments:

Line 575: Please define “AEL”

Line 176: Please define “PGC”

Line 182: I think the word “to” should be “from”

Line 268: What are the “appropriate humidity levels”? This information would be helpful for someone trying to use your methods.

Line 575: Please define “AEL” in this figure legend.

Line 639-641: Please add a citation for this statement abou nuclease resistance and reduced immune response.

Line 687: “indicate” should be “indicates”

Line 710: “Fundatrix” should be “Fundatrices”

Line 828: Tunning (typo)

Lines 890-891: Please mention how many control experiments were conducted with BCR3, to give context to the “close to 0%” value. The “close to” qualification is surprising, as I would expect 0%. I wonder why there were any at all.

Lines 1000-1016: having just read the results, this seems to simply repeat them. Perhaps it can be telescoped so that you can proceed more quickly to the discussion points?

Line 1112 “an another” (remove “an”)

Line 1122 CRISPR/as9 (typo)

**Have all data underlying the figures and results presented in the manuscript been provided?**

Reviewer #1: None

Reviewer #2: Yes

Reviewer #3: Yes

PLOS authors have the option to publish the peer review history of their article (what does this mean? ). If published, this will include your full peer review and any attached files.

**Do you want your identity to be public for this peer review?** For information about this choice, including consent withdrawal, please see our Privacy Policy .

Reviewer #1: No

Reviewer #2: No

Reviewer #3: No

**Figure resubmission:**
---

## [Decision Letter · Decision Letter 1]

Dear Dr Shigenobu,

We are pleased to inform you that your manuscript entitled "Refined CRISPR/Cas9 genome editing in the pea aphid uncovers the essential roles of Laccase2 in overwintering egg adaptation" has been editorially accepted for publication in PLOS Genetics. Congratulations!

Yours sincerely,

Subba Reddy Palli, Ph.D.

Academic Editor

PLOS Genetics

Quanjiang Ji

Section Editor

PLOS Genetics

Aimée Dudley

Editor-in-Chief

PLOS Genetics

Anne Goriely

Editor-in-Chief

PLOS Genetics

Comments from the reviewers (if applicable):

Please correct the missing arrowhead in Figure 5 in the proofs.

Reviewer's Responses to Questions

**Comments to the Authors:**

Reviewer #1: Please correct the missing arrowhead in Figure 5c. Otherwise, it is OK.

Reviewer #2: I appreciate the authors' efforts for revision. I did not have any further comments.

Reviewer #3: The revision is thorough and has addressed all of my concerns.

**Have all data underlying the figures and results presented in the manuscript been provided?**

Reviewer #1: None

Reviewer #2: None

Reviewer #3: Yes

PLOS authors have the option to publish the peer review history of their article (what does this mean? ). If published, this will include your full peer review and any attached files.

**Do you want your identity to be public for this peer review?** For information about this choice, including consent withdrawal, please see our Privacy Policy .

Reviewer #1: **Yes: ** Ryo Futahashi

Reviewer #2: No

Reviewer #3: No

**Data Deposition**

http://datadryad.org/submit?journalID=pgenetics&manu=PGENETICS-D-24-01533R1

**Press Queries**

---

## [Editor Report · Acceptance letter]

PGENETICS-D-24-01533R1

Refined CRISPR/Cas9 genome editing in the pea aphid uncovers the essential roles of Laccase2 in overwintering egg adaptation

Dear Dr Shigenobu,

We are pleased to inform you that your manuscript entitled "Refined CRISPR/Cas9 genome editing in the pea aphid uncovers the essential roles of Laccase2 in overwintering egg adaptation" has been formally accepted for publication in PLOS Genetics! Your manuscript is now with our production department and you will be notified of the publication date in due course.

With kind regards,

Lilla Horvath

PLOS Genetics

On behalf of:
